EMBO
Molecular Medicine

# S100A1 is released from ischemic cardiomyocytes and signals myocardial damage via Toll-like receptor 4

David Rohde[1], Christoph Schön[1], Melanie Boerries[2,3], Ieva Didrihsone[1], Julia Ritterhoff[1], Katharina F Kubatzky[4], Mirko Völkers[1], Nicole Herzog[1], Mona Mähler[1], James N Tsoporis[5], Thomas G Parker[5], Björn Linke[6], Evangelos Giannitsis[1], Erhe Gao[7], Karsten Peppel[7], Hugo A Katus[1,8] & Patrick Most[1,7,8,*]

## Abstract

Members of the S100 protein family have been reported to function as endogenous danger signals (alarmins) playing an active role in tissue inflammation and repair when released from necrotic cells. Here, we investigated the role of S100A1, the S100 isoform with highest abundance in cardiomyocytes, when released from damaged cardiomyocytes during myocardial infarction (MI). Patients with acute MI showed significantly increased S100A1 serum levels. Experimental MI in mice induced comparable S100A1 release. S100A1 internalization was observed in cardiac fibroblasts (CFs) adjacent to damaged cardiomyocytes. *In vitro* analyses revealed exclusive S100A1 endocytosis by CFs, followed by Toll-like receptor 4 (TLR4)-dependent activation of MAP kinases and NF-κB. CFs exposed to S100A1 assumed an immunomodulatory and anti-fibrotic phenotype characterized i.e. by enhanced intercellular adhesion molecule-1 (ICAM1) and decreased collagen levels. In mice, intracardiac S100A1 injection recapitulated these transcriptional changes. Moreover, antibody-mediated neutralization of S100A1 enlarged infarct size and worsened left ventricular functional performance post-MI. Our study demonstrates alarmin properties for S100A1 from necrotic cardiomyocytes. However, the potentially beneficial role of extracellular S100A1 in MI-related inflammation and repair warrants further investigation.

**Keywords** alarmin; cardiac fibroblast; damage-associated molecular pattern (DAMP); S100A1; Toll-like receptors (TLRs)
**Subject Categories** Cardiovascular System; Immunology

## Introduction

Acute myocardial infarction (MI) resulting in cardiomyocyte death entails the coordinated activation of a cytokine cascade that initiates an acute inflammatory reaction (Frangogiannis *et al*, 2002; Ertl & Frantz, 2005). However, the molecular triggers that actually link cardiomyocyte necrosis to initiation of the inflammatory phase are not well understood. In accordance with the danger model by Matzinger (2002), an emerging concept in cardiac tissue damage and healing is that intracellular molecules passively released from damaged cells can play an active role in the restoration of tissue homeostasis.

These molecules are referred to as alarmins or damage-associated molecular patterns (DAMP) and signal cellular damage to resident target cells via molecular pattern recognition receptors such as the receptor for advanced glycation end products (RAGE) and members of the Toll-like receptor family (TLR) (Chan *et al*, 2012). The alarmin family comprises a group of evolutionary unrelated and structurally diverse endogenous molecules with defined intracellular functions (Foell *et al*, 2007a). The growing list includes the prototypical high-mobility group box 1 protein (HMGB1) and members of the S100 protein family [reviewed in Bianchi, 2007; Chan *et al*, 2012; Foell *et al*, 2007a; Oppenheim & Yang, 2005]. Due to their diversity, alarmins generate specific cytokine signatures in their target cells that shape both quality and intensity of subsequent inflammatory phases (Klune *et al*, 2008).

HMGB1 is an ubiquitous non-histone nuclear protein involved in transcriptional control (Lotze & Tracey, 2005). It is rapidly released from necrotic tissue into the interstitial space (Scaffidi *et al*, 2002). In the injured heart, it can exert both pro-inflammatory and

1   Section of Molecular and Translational Cardiology, Department of Internal Medicine III, Heidelberg University Hospital, Heidelberg University, Heidelberg, Germany
2   Institute of Molecular Medicine and Cell Research, Freiburg University, Freiburg, Germany
3   German Consortium for Translational Cancer Research (DKTK), Partner site Freiburg, German Cancer Research Center (DKFZ), Heidelberg, Germany
4   Division for Microbiology and Hygiene, Department of Infectious Diseases, Heidelberg University Hospital, Heidelberg University, Heidelberg, Germany
5   Division of Cardiology, Department of Medicine, Keenan Research Centre, Li Ka Shing Knowledge Institute, St. Michael's Hospital, University of Toronto, Ontario, Canada
6   Division of Immunogenetics, Tumor Immunology Program, German Cancer Research Center (DKFZ), Heidelberg, Germany
7   Center for Translational Medicine, Department of Medicine, Thomas Jefferson University, Philadelphia, PA, USA
8   German Centre for Cardiovascular Research (DZHK), Partner site Heidelberg/Mannheim, Heidelberg University Hospital, Heidelberg University, Heidelberg, Germany
    *Corresponding author. Tel: +49 6221 568900; Fax: +49 6221 567632; E-mail: patrick.most@med.uni-heidelberg.de

pro-fibrotic actions through the activation of RAGE (Andrassy *et al*, 2008). In addition, HMGB1 directly suppresses cardiomyocyte contractility (Tzeng *et al*, 2008). Post-MI neutralization of HMGB1 release mitigated MI size and numerous other studies reported a detrimental role of HMGB1 in the pathogenesis of chronic inflammatory diseases (Lotze & Tracey, 2005; Tsung *et al*, 2005; Foell *et al*, 2007a; Andrassy *et al*, 2008). Similar characteristics were attributed to S100A8 and S100A9 in various inflammatory conditions including cardiac and vascular disease (Hirono *et al*, 2006; Foell *et al*, 2007b; Boyd *et al*, 2008). Both members of the multigenic S100 EF-hand $Ca^{2+}$ sensor protein family were previously identified as extracellular pro-inflammatory and pro-fibrotic mediators acting on various cardiac cell types including ventricular cardiomyocytes (VCMs) and cardiac fibroblasts (CFs) via TLR4 and RAGE (Ehlermann *et al*, 2006; Averill *et al*, 2012; Zhang *et al*, 2012).

The emerging role of S100 proteins as putative alarmins in the injured heart attracted interest in a potentially similar function of S100A1. It is the S100 isoform with highest abundance in cardiomyocytes (Kato & Kimura, 1985; Kiewitz *et al*, 2000; Brinks *et al*, 2011) where it acts as a $Ca^{2+}$-dependent molecular inotrope (Most *et al*, 2001). S100A1 regulates and improves function of key proteins involved in the control of sarcoplasmic reticulum $Ca^{2+}$ handling as well as myofilament and mitochondrial function, thereby enhancing the heart's inotropic and lusitropic state [reviewed in (Most *et al*, 2007; Ritterhoff & Most, 2012; Rohde *et al*, 2010)]. Most recently, Bi *et al* (2013) reported depletion of S100A1 in ischemic cardiomyocytes, suggesting a passive release from damaged cells. However, it is currently unknown whether extracellular S100A1 exerts biological effects in the injured heart.

Originating from clinical data showing S100A1 release in patients with acute MI, the study presented here is the first examining extracellular S100A1 as a cardiac alarmin. Our comprehensive molecular study details how extracellular S100A1 evokes a distinct immunomodulatory and anti-fibrotic phenotype transition in CFs. S100A1's extracellular actions rely on endocytosis and endolysosomal TLR4-dependent signaling via transient mitogen- and stress-activated protein kinases (MAPK/SAPK) and activation of the nuclear factor kappa B (NF-κB) component p65. Potential clinical relevance for ischemia-released S100A1 as a direct molecular link between cardiomyocyte death and beneficial post-MI healing emanates from the fact that neutralization of extracellular S100A1 enlarged MI size and enhanced pro-fibrotic marker expression. Hence, our mechanistic study advances our understanding of S100A1's function in the heart and prompts continued investigation of its novel role as cardiac alarmin in post-MI healing. In support of a cardiac danger model, our findings might bear potential for new immunomodulatory strategies aiming at improved infarct healing and repair.

# Results

## S100A1 is released into the circulation of patients and mice with acute ST-segment elevation myocardial infarction

Ischemic hearts release endogenous molecules from necrotic cardiomyocytes into the interstitial space and circulation (Frangogiannis *et al*, 2002). Whether the cardiomyocyte protein S100A1 is released

from ischemic hearts was assessed by enzyme-linked immunosorbent assay (ELISA) in patients with acute ST-segment elevation myocardial infarction (STEMI) (for detailed patients' characteristics, see Supplementary Fig S1A). Figure 1A shows the admission electrocardiogram of a representative STEMI patient with occlusion of the main left coronary artery. S100A1 serum concentrations of the same patient peaked approximately 8 h after the onset of clinical symptoms (Supplementary Fig S1B). S100A1 peak serum levels in the STEMI group, assessed 6–12 h after occurrence of chest pain, were significantly elevated in comparison with control patients not admitted for acute cardiac events (Fig 1B). Augmented high-sensitive troponin T (hsTnT) and creatine kinase (CK) levels confirmed myocardial necrosis (Supplementary Fig S1D–E). Surgically induced STEMI in C57Bl/6 wild-type (WT) mice by left anterior descending coronary artery (LAD) occlusion (Fig 1C) mirrored the transient rise in S100A1 serum concentrations (Fig 1D and Supplementary Fig S1C). This provided us with an experimental model to investigate a potential role for damage-released S100A1 as a cardiac signaling molecule.

## S100A1 released from ischemic cardiomyocytes in infarcted murine hearts is internalized by adjacent cardiac fibroblasts

We first determined the fate of liberated S100A1 protein within ischemic murine myocardium. Confocal immunofluorescent (IF) microscopy of sham-operated control hearts confirmed the previously described S100A1 striated pattern in ventricular cardiomyocytes (VCMs) (Fig 2A) (Most *et al*, 2004a). Neighboring interstitial cardiac fibroblasts (CFs) identified by the fibroblast-specific discoidin domain receptor 2 (DDR2) were devoid of S100A1 (Fig 2A and Supplementary Fig S2A). *In vitro* studies confirmed that cultured CFs isolated from control hearts neither possessed nor expressed S100A1 mRNA and protein (Supplementary Fig S5A). In contrast, DDR2-positive CFs adjacent to damaged VCMs in the ischemic border zone of infarcted hearts stained positive for intracellular S100A1 (Fig 2B and Supplementary Fig S2B–C). Consistent with this, enzymatically isolated non-cardiomyocyte fractions from ischemic hearts, which mainly consist of CFs, yielded enhanced S100A1 protein content compared with corresponding fractions of control hearts (Supplementary Fig S2D). On the other hand, S100A1 content of remote myocardium was comparable between sham-operated and infarcted hearts at 8, 24, and 48 h post-surgery, arguing against enhanced S100A1 expression in non-infarcted myocardium early post-MI (Supplementary Fig S2E). In further support of a cardiomyocyte origin, cultured CFs subjected to S100A1-containing supernatant from necrotic VCMs showed intracellular positive staining for S100A1 *in vitro* (Fig 2C). This result corroborates the notion that S100A1 released from ischemic cardiomyocytes *in vivo* is internalized by interstitial adjacent CFs.

## Endocytosed S100A1 is delivered to acidic endolysosomes in cardiac fibroblasts

The question whether S100A1 internalization might exclusively occur in CFs was addressed *in vitro* in cultures of isolated permanent myocardial cell populations using rhodamine-labeled human recombinant S100A1 (rho-S100A1). Confocal IF analyses of CFs subjected to rho-S100A1 revealed rapid vesicular-like intracellular

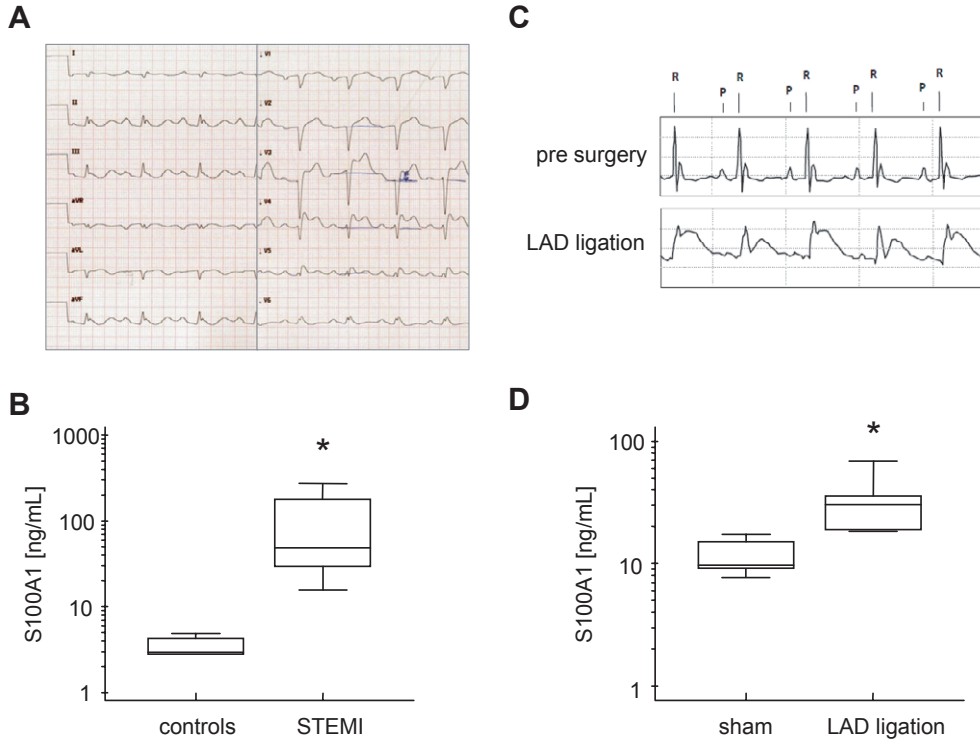

**Figure 1.  S100A1 release from ischemic human and murine myocardium.**

A    Admission 12-lead ECG from a patient with acute ST-segment elevation myocardial infarction (STEMI).

B    Average S100A1 serum peak concentrations in control ($n = 12$) and STEMI ($n = 12$) patients. Control patients presented to the internal medicine emergency department without acute cardiac events (*$P = 0.004$; see Supplementary Fig S1 for patients' characteristics).

C    Representative ECG recording of a C57Bl/6 mouse prior to (upper panel) and after surgical occlusion of the left anterior descending coronary artery (LAD) (lower panel) showing ST-segment elevation.

D    Average S100A1 serum concentrations 8 h after sham surgery ($n = 10$) and LAD ligation ($n = 10$) sampled by ELISA (*$P = 0.007$).

absorption of the fluorescent molecule, suggesting an endocytotic process (Fig 3A, upper panel). Pre-treatment with LPS or HMGB1 had no effect on the amount of endocytosed S100A1 (Supplementary Fig S5B). In contrast, rho-S100A1 internalization could neither be detected in isolated adult VCMs (Fig 3A, lower panel) nor in smooth muscle (SMCs) or endothelial cells (ECs) (Supplementary Fig S3A). Co-localization studies further detailed delivery of rho-S100A1 to acidic endolysosomes in CFs using a fluorescence-labeled specific organelle marker (Fig 3B, upper panel). Intracellular signal separation between rho-S100A1 and fluorescence-labeled mitochondria (Fig 3B, lower panel) confirmed the specificity of our finding. Indicative of exclusive endocytosis into CFs, we next sought to characterize underlying mechanisms mediating S100A1 internalization.

**S100A1 endolysosomal trafficking relies on Toll-like receptor 4**

Since routing to endolysosomes suggests receptor-mediated uptake and trafficking (Doherty & McMahon, 2009), we studied rho-S100A1 uptake in isolated fibroblasts either deficient for the alarmin cell surface receptor RAGE ($RAGE^{-/-}$) or TLR4 ($TLR4^{-/-}$). Alike WT control cells, both $RAGE^{-/-}$ and $TLR4^{-/-}$ fibroblasts internalized rho-S100A1 (Fig 4A and Supplementary Fig S5C). However, only TLR4-competent fibroblasts (WT and $RAGE^{-/-}$)

directed S100A1 to acidic endolysosomes (Fig 4A, see Supplementary Fig S4 for additional images and statistical analysis). Co-localization of S100A1 with TLR4 in CFs using a proximity ligation assay further substantiated the notion of S100A1 binding to TLR4 located within endolysosomes (Supplementary Fig S3D). To determine whether intracellular S100A1 trafficking requires the TLR adaptor myeloid differentiation factor-88 (MyD88) or relies on the TLR4 ligand-binding ectodomain only, we studied rho-S100A1 subcellular location in MyD88-knockout fibroblasts ($MyD88^{-/-}$). The latter still showed S100A1 routing to the acidic endosomal compartment (Fig 4A, lowest panel). Further consolidating a cell surface receptor-independent uptake of rho-S100A1, usage of IF markers as well as chemical (Fig 4B and C) and genetic inhibitors (Supplementary Fig S3B) either sensitive to clathrin- or caveolin-mediated endocytosis showed unchanged rho-S100A1 uptake in CFs compared to controls. Interestingly, pre-incubation with macropinocytosis inhibitor amiloride resulted in abolished rho-S100A1 internalization (Supplementary Fig S3C). Moreover, treatment of WT fibroblasts with Rac1-inhibitor EHT1864 prevented S100A1 endolysosomal trafficking (Supplementary Fig S4B). These results both suggest RAGE- and TLR4-independent internalization of S100A1 most likely due to fluid endocytosis but point toward TLR4-Rac1-dependent trafficking to an endolysosomal TLR4/MyD88 signaling complex.

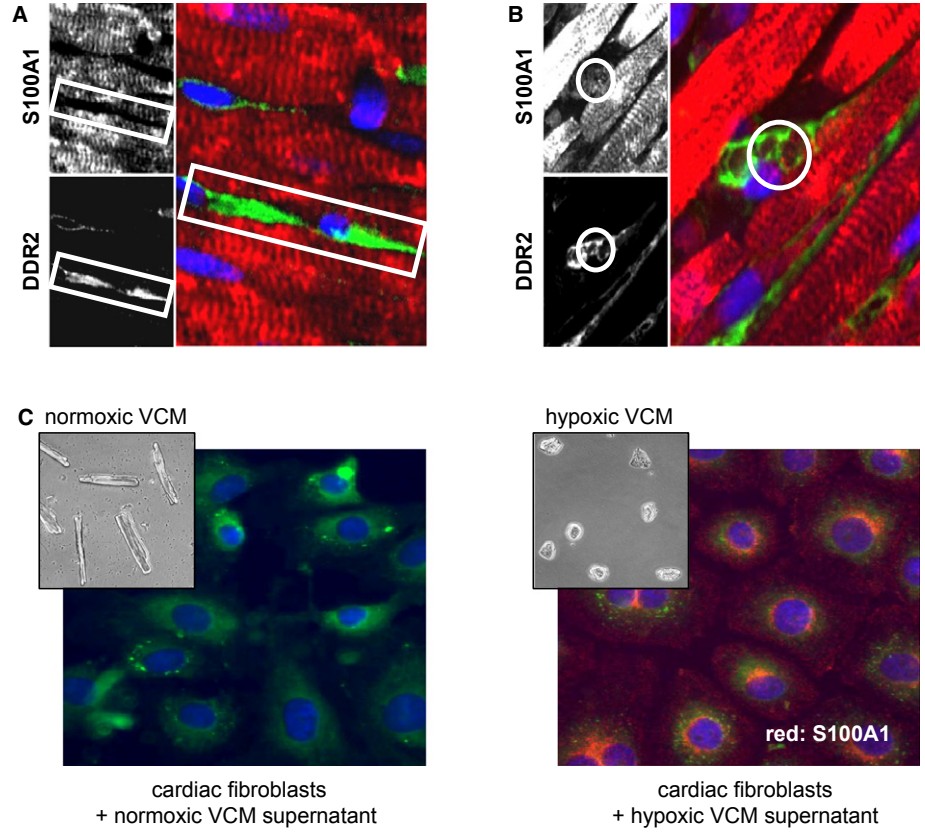

**Figure 2.  S100A1 released from ischemic cardiomyocytes is internalized by cardiac fibroblasts *in vivo* and *in vitro*.**

A   Representative immunofluorescent (IF) staining of sham-operated murine myocardium 48 h after surgery (red and upper grayscale image: S100A1; green and lower grayscale image: discoidin domain receptor 2/DDR2; blue: DAPI/nuclei). The white box highlights the S100A1-negative space congruent with DDR2-positive cardiac fibroblasts (CFs).

B   Representative IF staining of the border zone of infarcted murine myocardium 48 h after LAD ligation (red: S100A1, green: DDR2). DDR2-labeled CFs adjacent to cardiomyocytes are positive for S100A1 (white circle: S100A1 accumulation superimposable with DDR2-positive CF) (magnification 40×; see Supplementary Fig S2 for more images).

C   CFs were subjected to medium taken from normoxic cardiomyocytes (VCM, left brightfield image) and hypoxic cardiomyocytes (right brightfield image). Representative IF staining shows intracellular S100A1 staining (red) only in CFs incubated with hypoxic cardiomyocyte supernatant (right IF image) (green: endoplasmic reticulum, blue: DAPI; magnification 40×).

## Internalized S100A1 transiently activates MAPK/SAPK and NF-κB signaling

Uptake of extracellular S100A1 protein resulted in a transient time- and dose-dependent activation of ERK1/2, p38, and SAPK/JNK in CFs as assessed by phospho-specific Western blot analyses (Fig 5A and Supplementary Fig S6A). Electrophoretic mobility shift assay and IF techniques revealed a similar activation course for NF-κB transcription factor p65 and enhanced nuclear import of phosphorylated p65 (Fig 5B–D). On the contrary, endogenous molecules released from necrotic cardiomyocytes such as CK or TnT as well as other EF-hand $Ca^{2+}$-binding proteins including S100A4 and calmodulin neither elicited MAPK/SAPK nor p65 activation in CFs when administered as recombinant factors (Supplementary Fig S6B). Control experiments confirmed that MAPK/SAPK and p65 activation was not due to the contamination of recombinantly synthesized S100A1 with endotoxin (Supplementary Fig S6C). Specificity of S100A1-mediated signaling was further strengthened by absent

activation of Akt, STAT3, and unchanged reactive oxygen species (ROS) production in CFs (Supplementary Fig S6D and E). In line with our *in vitro* uptake studies, extracellular S100A1 neither evoked changes in MAPK/SAPK nor p65 activity in isolated VCMs, SMCs, or ECs.

## S100A1 signaling in cardiac fibroblasts occurs through endosomal TLR4/MyD88

Usage of WT, $RAGE^{-/-}$, $TLR4^{-/-}$, and $MyD88^{-/-}$ fibroblasts unveiled that S100A1 activation of MAPK/SAPK and p65 is independent of RAGE but requires both TLR4 and its adaptor protein MyD88 (Fig 5E). Preserved specificity of TLR-dependent signaling in $RAGE^{-/-}$, $TLR4^{-/-}$, and $MyD88^{-/-}$ fibroblasts was confirmed by regular activation patterns induced by the MyD88-dependent TLR4 and TLR1/2 agonists LPS-EB and Pam3, respectively, as well as MyD88-independent TLR3 ligand Poly I:C (Fig 5E). Inhibition of S100A1-mediated ERK1/2 activation by chloroquine, which

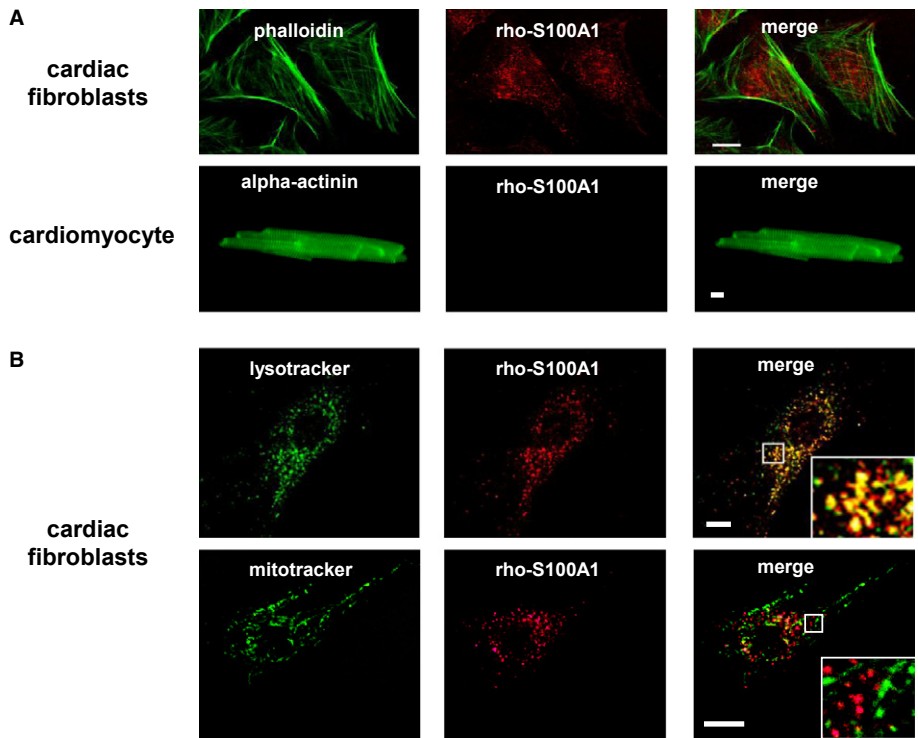

**Figure 3.　S100A1 uptake is cardiac fibroblast specific and routed to the endolysosomal compartment.**

A　Representative immunofluorescence (IF) image of cardiac fibroblasts (CFs, upper panel) and an adult cardiomyocyte (lower panel) subjected to 1 μM rhodamine-labeled S100A1 protein (rho-S100A1, red). Counterstaining with FITC-labeled phalloidin and anti-alpha actinin (both green), respectively. Adult cardiomyocytes show no internalization of S100A1 (see Supplementary Fig S3 for smooth muscle cells and endothelial cells) (scale bar, 10 μm).

B　Representative IF images of CFs revealing substantial co-localization of absorbed rho-S100A1 (red) and FITC-labeled lysotracker (green) which accumulates in acidified endolysosomes (upper panel). Staining with FITC-labeled mitotracker (green) displays disparate subcellular locations of internalized S100A1 (red) and mitochondria (lower panel) (scale bars, 10 μm).

prevents endosomal acidification, indicated that S100A1 signaling might originate from a TLR4/MyD88 endolysosomal complex (Fig 5F). Preserved p65 phosphorylation in the presence of the MEK1/2 inhibitor PD98059 suggested concurrent but independent activation of MAPK and NF-κB pathways (Supplementary Fig S6F). Overall, the biochemical results suggest that transient ischemic S100A1 release from necrotic cardiomyocytes induces temporary MAPK/SAPK and p65 signaling in CFs through TLR4/MyD88. Given the impact of these signaling pathways on CF function, we next investigated the effect of S100A1 internalization on the cellular phenotype.

**Endocytosed S100A1 evokes an immunomodulatory phenotype in cardiac fibroblasts**

Due to the ability of alarmins to trigger cytokine, chemokine, and cell adhesion molecule expression in target cells (Chan *et al*, 2012), we further determined expression changes in immunomodulatory factors. Transcript levels of intercellular adhesion molecule 1 (ICAM1), interleukin-10 (IL-10), thrombospondin-2 (TSP-2), and stromal cell-derived factor 1 (SDF1) were significantly changed in response to extracellular S100A1 (Fig 6A), in addition to a larger number of immunomodulatory genes (Supplementary Fig S7B). Subsequent protein elevations in ICAM1, TSP-2, and IL-10 were

identified by Western blotting and ELISA (Fig 6B-C). In line with our signaling analyses, usage of WT, $RAGE^{-/-}$, $TLR4^{-/-}$, and $MyD88^{-/-}$ fibroblasts demonstrated that S100A1-mediated expression changes in ICAM1 are independent of RAGE but require both TLR4 and MyD88 (Supplementary Fig S8A). Ensuing experiments with the chemical NF-κB and ERK1/2 signaling inhibitors MG-132 and PD98059, respectively, linked S100A1-mediated p65 and MAPK activation to expression changes both of immunomodulatory and ECM modulatory factors as exemplified by ICAM1, TSP-2, or MMP9 (Supplementary Fig S8B). Cell counts and [³H]-thymidine incorporation studies revealed that extracellular S100A1 did not affect CF proliferation or viability (Supplementary Fig S8C and D).

**Internalized S100A1 provokes an anti-fibrotic phenotype transition in cardiac fibroblasts**

In light of previously reported pro-fibrotic actions of released alarmins from necrotic cardiomyocytes such as HMGB1, S100A8, and S100A9 (Zhang *et al*, 2012), we next determined the impact of extracellular S100A1 on CF-derived extracellular matrix (ECM) constituents and modulatory factor expression. In response to extracellular S100A1, diminished collagen type 1 (col-1) mRNA levels accompanied by enhanced matrix metalloproteinase 9 (MMP9) expression were found (Fig 6D). In addition, mRNA levels of the

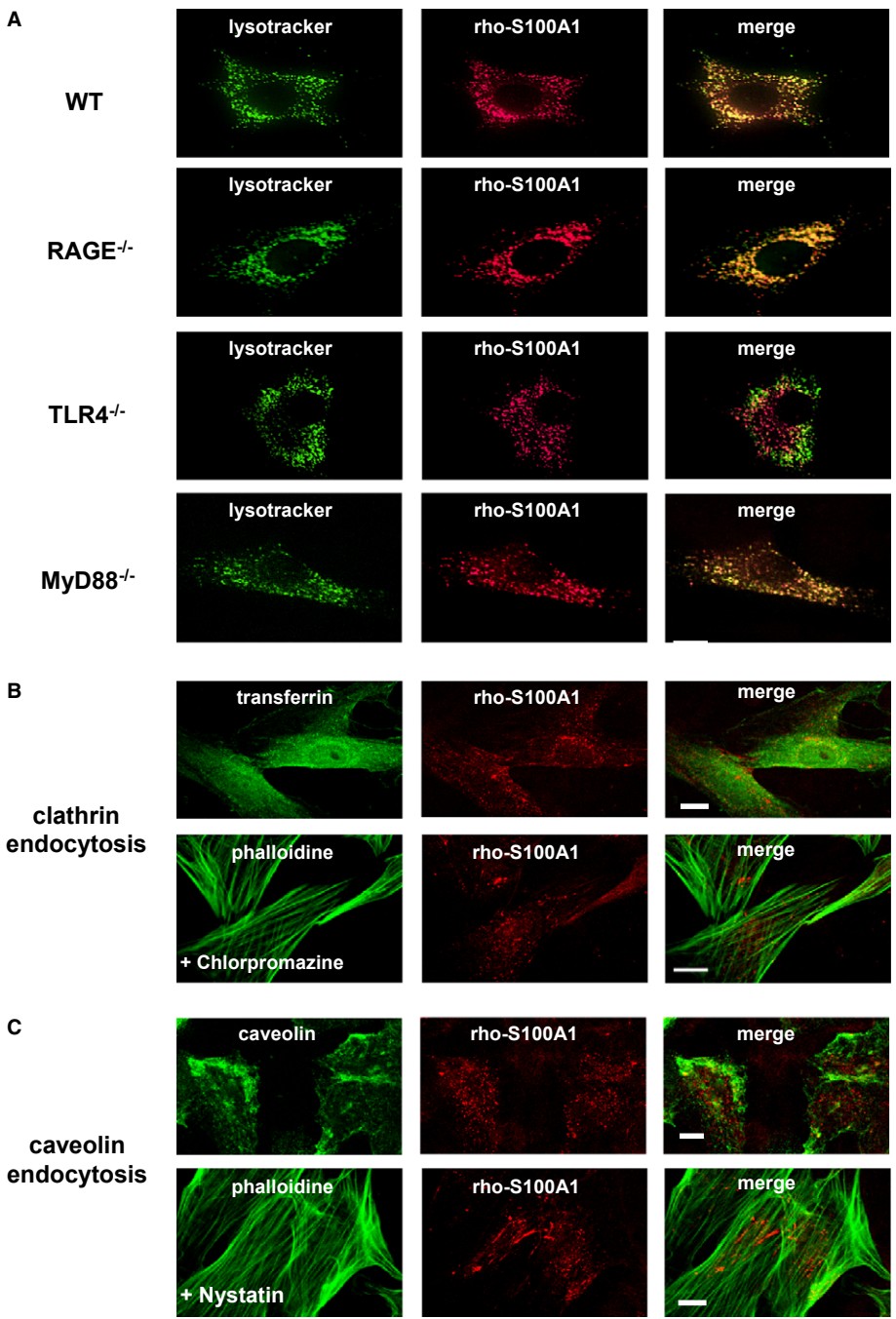

**Figure 4. S100A1 endolysosomal trafficking is TLR4 dependent, whereas S100A1 internalization is independent of clathrin and caveolin.**

A   Representative IF stainings of rho-S100A1 internalization (1 μM, red) in wild-type (WT), $RAGE^{-/-}$, $TLR4^{-/-}$, and $MyD88^{-/-}$ fibroblasts. Merge of images reveals missing co-localization of S100A1 with FITC-lysotracker (green) only in $TLR4^{-/-}$ cells (scale bar, 10 μm).

B   IF staining of cardiac fibroblasts (CFs) for clathrin-coated pits (transferrin, green) and internalized rho-S100A1 (1 μM, red) showing disparate subcellular locations (upper panel). Pre-treatment with chlorpromazine (10 μM), a chemical inhibitor of clathrin-mediated endocytosis, did not prevent rho-S100A1 absorption (counterstaining with FITC-phalloidin, green) (lower panel) (scale bars, 10 μm).

C   IF staining of CFs for caveolin-1 (green) and rho-S100A1 (red) showing no co-localization (upper panel). Pre-treatment with nystatin (10 μM), a chemical inhibitor of caveolin-mediated endocytosis, did not prevent rho-S100A1 uptake (lower panel) (scale bars, 10 μm).

myofibroblast marker and inducing factor smooth muscle actin (SMA) and connective tissue growth factor (CTGF), respectively, were also decreased (Fig 6D). Supplementary Fig S7A demonstrates the time- and dose-dependent attenuation of col-1 mRNA by S100A1. In contrast, other ECM factors such as fibronectin or tissue inhibitors of metalloproteinase 1 and 2 (TIMP1/2) were unchanged

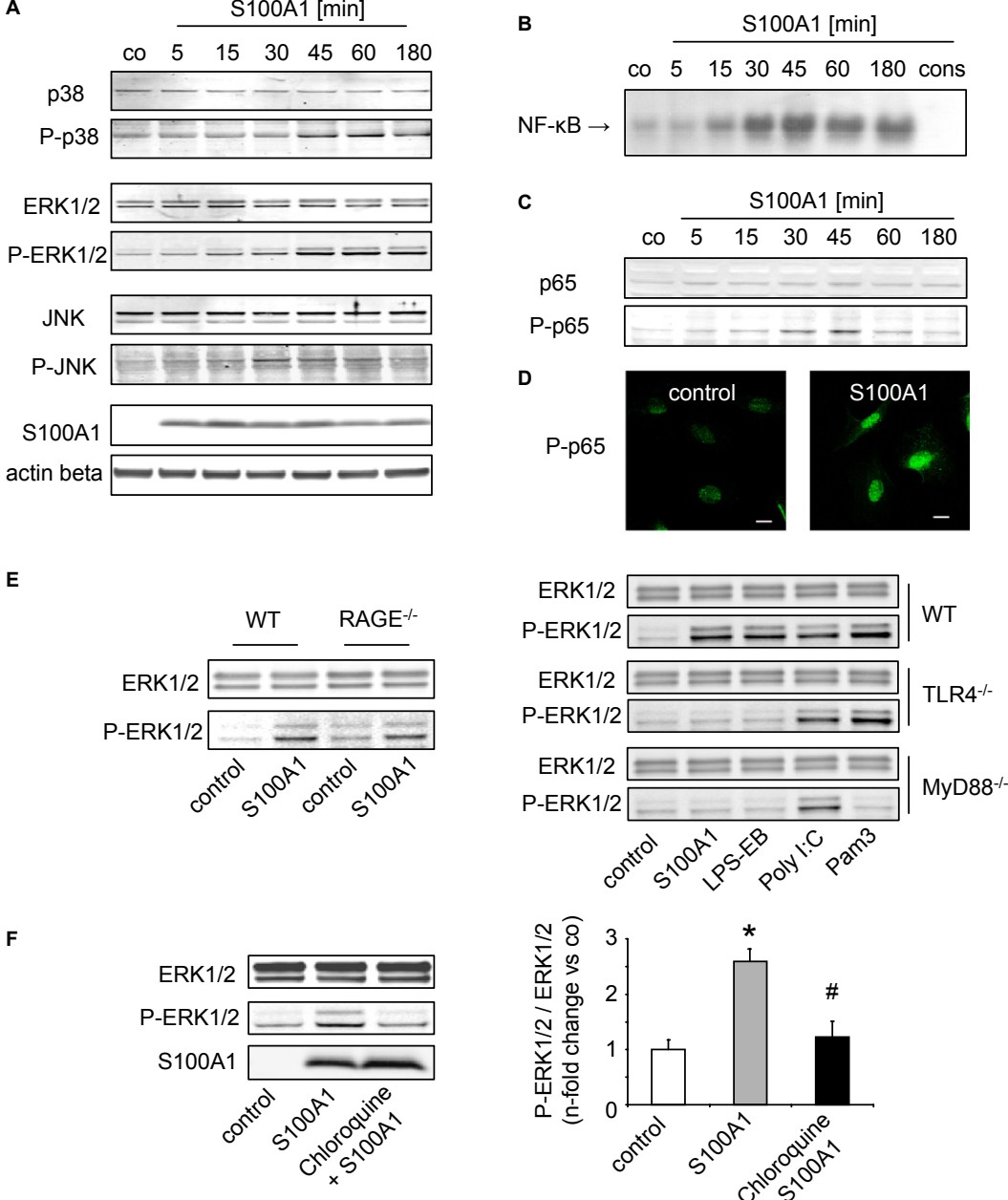

**Figure 5. Endocytosed S100A1 activates MAPK/SAPK and NF-κB signaling via a TLR4/MyD88-dependent mechanism.**

A    Representative Western blots (WBs) demonstrating time-dependent transient ERK1/2, p38, and JNK activation in cardiac fibroblasts (CFs) by 1 μM extracellular S100A1 (see Supplementary Fig S6A for dose titration).

B, C    NF-κB and p65, assessed by electrophoretic mobility shift assay (B) and WB (C), show a similar activation course in response to S100A1.

D    IF images depicting nuclear import of phospho-p65 in S100A1-treated CFs (45 min).

E    WBs showing S100A1-mediated ERK1/2 activation only in WT and *RAGE*$^{-/-}$, but not in *TLR4*$^{-/-}$ and *MyD88*$^{-/-}$ fibroblasts. Regular TLR4/MyD88 signaling is confirmed by ERK1/2 activation in response to TLR4-ligand LPS-EB (MyD88 dependent), TLR3-ligand Poly I:C (MyD88 independent), and TLR1/2-ligand Pam3 (MyD88 dependent).

F    Representative WBs of CFs treated with chloroquine (10 μM), an inhibitor of endosomal acidification, showing abrogated S100A1-mediated ERK1/2 activation ($n = 3$ individual experiments, $*P = 0.02$ vs control, $^{\#}P = 0.03$ vs S100A1).

Source data are available for this figure.

(see Supplementary Fig S7B for further ECM genes affected by S100A1). Decreased collagen synthesis measured via $^{3}$H-hydroxy-proline incorporation and enhanced zymographic MMP9 activity

further substantiated the phenotypic change (Fig 6E-F). In line with aforementioned results, extracellular S100A1 did not affect VCM contractility or susceptibility to apoptosis, strengthening the

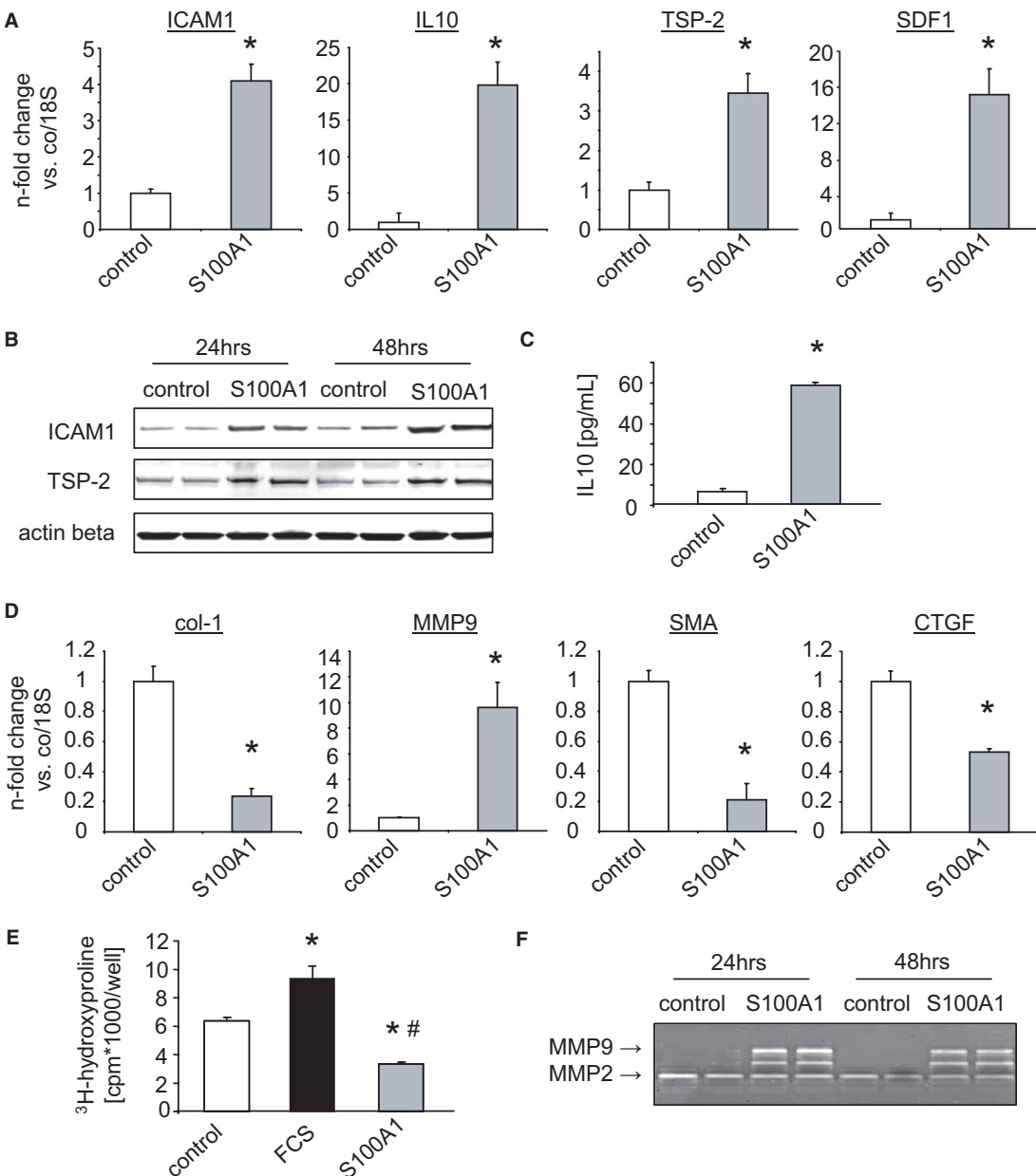

**Figure 6. Internalized S100A1 conveys an immunomodulatory and anti-fibrotic phenotype transition in cardiac fibroblasts.**

A   Measurement of mRNA level changes in intercellular adhesion molecule 1 (ICAM1), interleukin-10 (IL-10), thrombospondin-2 (TSP-2), and stromal cell-derived factor 1 (SDF1) in CFs treated with S100A1 (n = 5 individual experiments, *P-values vs control: 0.01 for ICAM1, 0.004 for IL-10, 0.02 for TSP-2, 0.03 for SDF1).

B   Western blots showing increased ICAM1 and TSP-2 protein expression in homogenates of S100A1-treated CFs.

C   Increased IL-10 concentration in the supernatants from CFs incubated with S100A1 as assessed by ELISA (n = 3 individual experiments, *P = 0.001 vs control).

D   Semi-quantitative assessment of transcriptional changes in collagen type 1 (col-1), matrix metalloproteinase 9 (MMP9), smooth muscle actin (SMA), and connective tissue growth factor (CTGF) in S100A1-treated cardiac fibroblasts (CFs) (n = 5 individual experiments, *P-values vs control: 0.02 for col-1, 0.003 for MMP9, 0.03 for SMA, 0.04 for CTGF).

E   $^3$H-hydroxyproline incorporation depicts decreased collagen synthesis in CFs in response to extracellular S100A1. FCS served as stimulatory control (n = 3 individual experiments, *P-values vs control: 0.02 for FCS, 0.03 for S100A1, #P = 0.02 vs FCS).

F   Representative in-gel zymography showing enhanced MMP9 but unchanged MMP2 activity in S100A1-treated CFs.

Source data are available for this figure.

specificity of its actions on CFs (Supplementary Fig S7C–D). These results indicate that extracellular S100A1 might convey a potent anti-fibrotic effect by changing the CF phenotype. Having shown that CFs can adopt features of a cardiac sentinel cell in response to extracellular S100A1 (Smith *et al*, 1997), we were determined to assess *in vivo* consequences of post-MI S100A1 release in the injured heart.

**S100A1 injection and neutralization of ischemia-released S100A1 modulate gene expression and myocardial infarct healing in mice**

Intramyocardial injections of S100A1 protein into the left ventricular (LV) anterior wall of normal C57Bl/6 mouse hearts via a small left intercostal thoracotomy were used to determine potential effects of extracellular S100A1 protein on CF gene expression

*in vivo*. Figure 7A shows that S100A1 protein injection resulted in altered expression of col-1, ICAM1, TSP-2, and MMP9 as well as IL-10 and SDF-1 that mimicked the immunomodulatory and anti-fibrotic transcriptional changes in CFs. In view of a congruent *in vitro* and *in vivo* biological activity of extracellular S100A1, C57Bl/6 mice were pre-treated either with unspecific IgG fractions or with an affinity-purified S100A1-neutralizing IgG antibody.

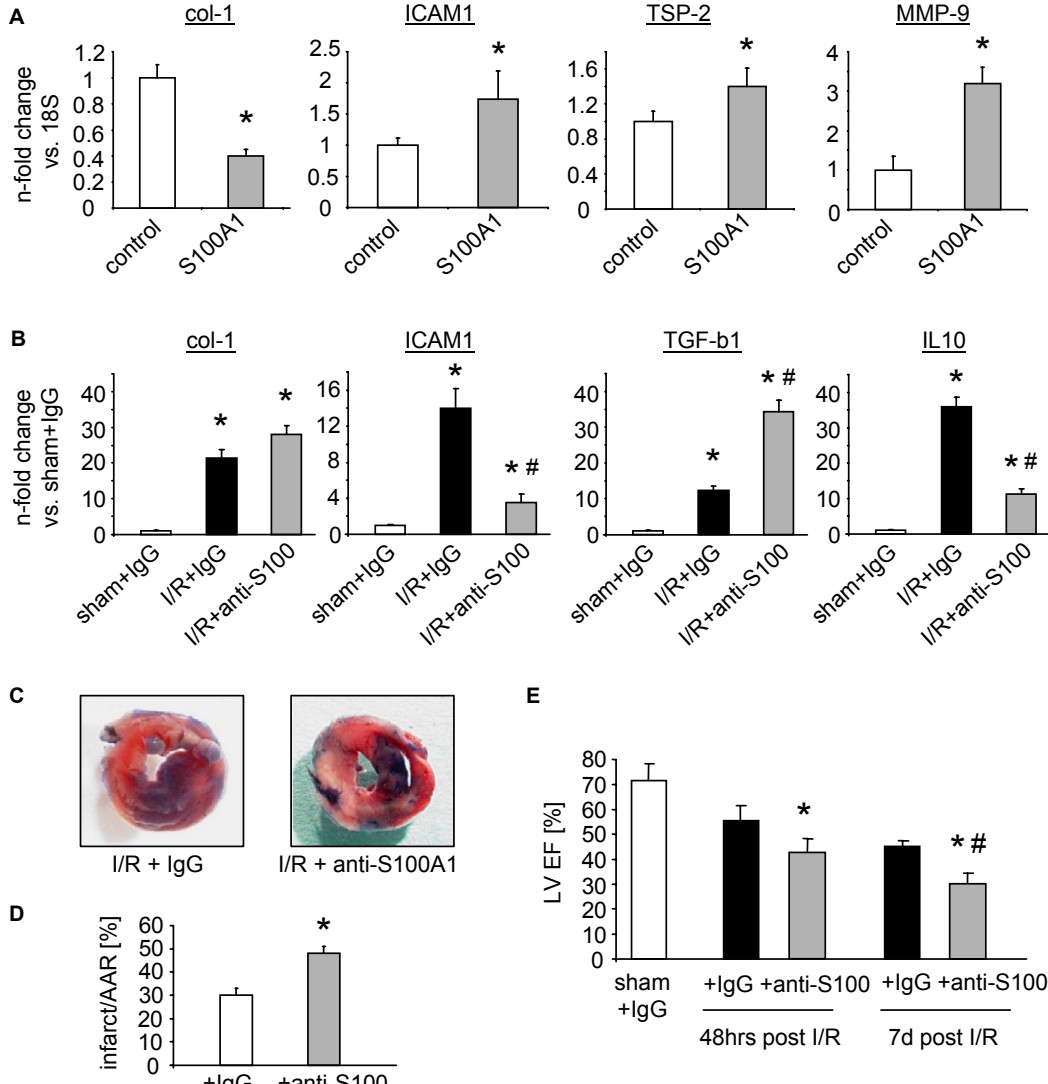

**Figure 7. Effects of S100A1 injection and neutralization on myocardial gene expression, infarct size, and functional performance *in vivo*.**

A   Changes in collagen type 1 (col-1), intercellular adhesion molecule 1 (ICAM1), thrombospondin-2 (TSP-2), and matrix metalloproteinase 9 (MMP9) mRNA levels after the injection of S100A1 protein into the left ventricular apical region of healthy mice (*n* = 4 animals per group, 3 injections in each heart, *P*-values vs control: 0.03 for col-1, 0.04 for ICAM1, 0.04 for TSP-2, 0.02 for MMP-9), mirroring the previously observed *in vitro* results.
B   Measurement of gene expression changes in col-1, transforming growth factor beta 1 (TGF-β1), ICAM1, and interleukin-10 (IL-10) mRNA levels in sham and infarcted murine myocardium pre-treated with either unspecific IgG or anti-S100A1 antibody by intraperitoneal injections (*n* = 8 animals per group, *P*-values vs sham+IgG: 0.003 and 0.01 for col-1, 0.001 and 0.02 for ICAM1, 0.03 and 0.01 for TGF-β1, 0.001 and 0.002 for IL-10, #*P*-values vs I/R+IgG: 0.01 for ICAM1, 0.03 for TGF-β1, 0.01 for IL-10). Anti-S100A1 pre-treatment resulted in exaggerated expression of col-1 and TGF-β1 together with abrogated ICAM1 and IL-10 upregulation.
C   Representative images of infarcted hearts stained with TTC and ethylene blue derived from control IgG and anti-S100A1-pre-treated C57Bl/6 mice.
D   Statistical analysis of enlarged infarct size in anti-S100A1-treated mice compared to IgG controls (*n* = 6 animals per group, *P* = 0.02 vs IgG).
E   Time course of post-infarction cardiac left ventricular function assessed by echocardiography showing greater deterioration of cardiac performance in anti-S100A1-treated animals (*P*-values vs corresponding IgG: 0.04 at 48 h, 0.03 at 7 days, #*P* = 0.03 vs anti-S100A1 48 h, *n* = 8 animals per group).

Intraperitoneal injection followed by experimental I/R injury due to temporary LAD ligation was carried out as described previously (Kuwahara *et al*, 2002). Ability of the affinity-purified anti-S100A1 antibody to neutralize S100A1 was confirmed *in vitro* using the supernatant of hypoxic VCMs (Supplementary Fig S9A). Of note, early changes (3 h post-I/R) in immunomodulatory gene expression such as ICAM1 and IL-10 were abrogated in infarcted mouse hearts after anti-S100A1 treatment (Fig 7B). On the contrary, post-MI pro-fibrotic marker expression such as col-1 and transforming growth factor beta 1 (TGF-β1) was significantly enhanced in anti-S100A1-pre-treated mice compared with controls. Similar results were observed in CFs *in vitro* that were treated with the supernatant from necrotic VCMs pre-incubated with the neutralizing anti-S100A1 antibody (Supplementary Fig S9B). Analysis of myocardial gene expression at time points from 3 h to 7 days post-I/R revealed prolonged myocardial inflammation in the anti-S100A1-treated group *in vivo* (Supplementary Fig S10). We finally assessed the overall impact of *in vivo* S100A1 neutralization on post-MI healing. Fig 7C-E shows that neutralization of S100A1 results in significantly increased MI size and impaired post-MI LV function when compared with IgG control injections. These results point toward a beneficial net effect of extracellular S100A1 in post-MI healing.

## Discussion

Our study is the first to describe a novel function of extracellular S100A1 as a cardiomyocyte-derived alarmin and uncovers the molecular mechanisms how S100A1 conveys an early immunomodulatory and anti-fibrotic effect in injured hearts. Like other types of sterile organ damage, ischemic myocardium releases endogenous cardiomyocyte moieties into the interstitial space and circulation (Frangogiannis *et al*, 2002). Due to the heart's cytoarchitecture, interstitial CFs that envelope cardiomyocytes with their extensions are most likely immanent recipients for these molecules (Souders *et al*, 2009). Besides their role as source of structural elements, fibroblasts have been identified as key sites of immunomodulatory factors involved in wound healing (Smith *et al*, 1997). The most salient findings of our study are that S100A1 released from damaged cardiomyocytes *in vivo* specifically targeted CFs and triggered transient TLR4-endolysosomal signaling. Downstream MAPK/SAPK and p65 activation conveyed an S100A1-mediated immunomodulatory and anti-fibrotic CF phenotype transition (Fig 8). The beneficial net effect on post-MI healing in mice suggests potential *in vivo* relevance for extracellular S100A1.

Clinical origin for this study was the characterization of cardiac ischemia in humans and mice as pathological condition that entails a significant release of S100A1 from injured cardiomyocytes. A previous study by Usui *et al* (1990) supports our clinical results, unveiling similar basal serum levels and time course of S100A1 in patients with acute MI. The same group reported that rapid clearance of S100A1 from the bloodstream occurs mainly through renal excretion due to the small molecular weight of the protein (Usui *et al*, 1989). Most recently, Bi *et al* (2013) showed depletion of S100A1 in ischemic cardiomyocytes, starting as early as 15 min after LAD ligation. Mimicking the transient release of S100A1 in a murine experimental MI model enabled first mechanistic

investigation of the fate of extracellular S100A1. Our *in vivo* data obtained in ischemic murine hearts indicate that S100A1 does not simply leave the interstitial space but is in part internalized by CFs adjacent to damaged cardiomyocytes.

This notion is supported by our systematic *in vitro* uptake analyses unveiling that neither adult cardiomyocytes nor SMCs or ECs internalize extracellular S100A1 as strongly as CFs. S100A1 uptake despite RAGE and TLR4 deficiency and various means to inhibit either clathrin- or caveolin-mediated endocytosis in CFs argues in favor of a receptor-independent uptake from the interstice. High fluid endocytotic activity distinguishes CFs from other cardiac cell types, rendering this mechanism likely for S100A1 internalization (Doherty & McMahon, 2009). CFs of control hearts were devoid of S100A1 arguing against exchange of S100A1 between intact cardiomyocytes and CFs. Since *in vitro* data excluded ischemic S100A1 production in CFs, our results support the view that S100A1 is passively released from damaged cardiomyocytes and subsequently internalized by adjacent CFs.

Previously estimated intracellular S100A1 concentrations in adult ventricular cardiomyocytes were in the micromolar range (Kato & Kimura, 1985; Kato *et al*, 1986). Upon ischemic cardiomyocyte damage, interstitial CFs are most likely exposed to approximate S100A1 concentrations *in vivo*. With respect to resulting biological activities, extracellular S100A1 seemed to differ fundamentally from previously described actions of alarmins like HMGB1 and S100A8 and S100A9 (Rock & Kono, 2008). These molecules were most recently reported to be released from damaged cardiomyocytes and stimulate CF proliferation together with pro-inflammatory and pro-fibrotic actions (Zhang *et al*, 2012). In addition, they suppress cardiomyocyte contractility in an autocrine manner and alter EC and SMC function (Ehlermann *et al*, 2006; Boyd *et al*, 2008; Tzeng *et al*, 2008; Yang *et al*, 2012). In contrast, extracellular S100A1 exclusively targets CFs function, leaving cardiomyocytes as well as SMCs and ECs unaffected. In particular, extracellular S100A1 did not suppress cardiomyocyte contractility.

Given a lack of systematic evaluation of myocardial cytokine and chemokine production by HMBG1, S100A8, and S100A9, it is difficult to directly compare their inflammatory actions with S100A1. The S100A1-induced cytokine and chemokine pattern in CFs, however, includes an almost evenly distributed number of anti-inflammatory and pro-inflammatory mediators. S100A1 strongly evoked, that is, the production of anti-inflammatory IL-10 as well as the stem cell recruiting chemokine SDF-1 (Banchereau *et al*, 2012; Penn *et al*, 2012), whereas TGF-β1 was downregulated by S100A1. In addition, S100A1 induced expression of TSP-2 that together with TSP-1 is known for their effect to contain inflammatory activity to the site of injury (Frangogiannis *et al*, 2005). Although S100A1's *in vivo* actions cannot simply be pinpointed to a single cytokine, chemokine, or a matricellular factor, our results support the notion of a balanced immunomodulatory rather than a polarized pro-inflammatory phenotype. Extracellular S100A1 may therefore attenuate early inflammatory actions and facilitate their timely resolution *in vivo*.

In contrast to pro-fibrotic actions of HMGB1, S100A8, and S100A9 due to expression induction of ECM components such as collagens, promotion of myofibroblast transition including SMA generation, and finally CF proliferation (Zhang *et al*, 2012), S100A1 exerts a time- and dose-dependent anti-fibrotic profile. Besides

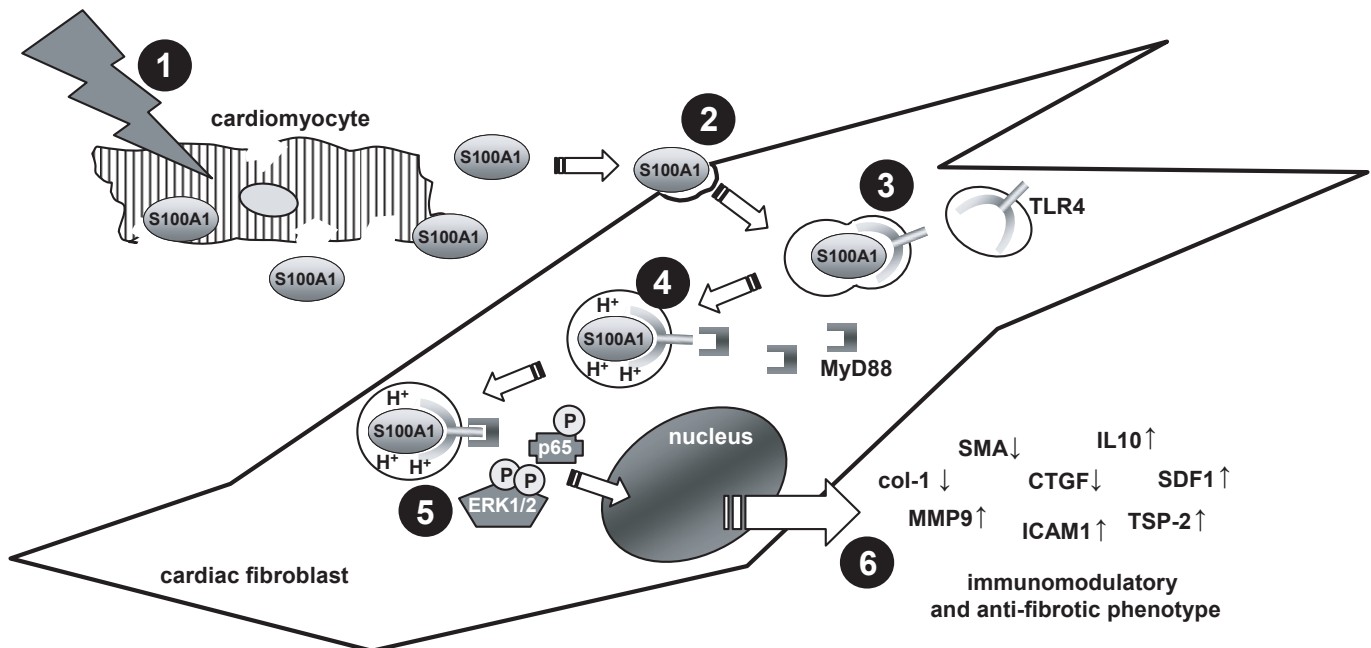

**Figure 8. Simplified scheme illustrating S100A1's role in myocardial infarction.**
1) Ischemic injury leads to the release of S100A1 from cardiomyocytes into the myocardial interstitium. 2) Extracellular S100A1 is specifically internalized by cardiac fibroblasts through clathrin- and caveolin-independent fluid endocytosis. 3) Endocytosed S100A1 binds to intracellular Toll-like receptor 4 (TLR4), thereby prompting trafficking to acidified endolysosomes. 4) MyD88 binds to the cytoplasmic domain of acidified TLR4 and initiates signal transduction. 5) The S100A1-TLR4-MyD88 signaling complex transiently activates MAPK/SAPK and NF-κB pathways. 6) S100A1-mediated signal transduction leads to an anti-fibrotic and immunomodulatory phenotype transition of cardiac fibroblasts. Neutralization of S100A1 resulted in significantly increased infarct size and impaired left ventricular function, suggesting a beneficial net effect of S100A1 in myocardial infarction.

distinct effects on numerous ECM components, extracellular S100A1 diminished the expression of collagens, inhibited the expression of the myofibroblast marker and inducer SMA and CTGF, respectively, and left CF proliferation unaffected. Given concomitant release of various alarmins from damaged cardiomyocytes (Zhang *et al*, 2012), it is tempting to speculate that S100A1 liberation might mitigate actions of pro-fibrotic alarmins, thereby preventing exaggerated post-MI fibrosis. Interestingly, extracellular CK-MB and TnT as well as CaM and S100A4 yielded no effect on CF activity, consolidating the specificity of S100A1 actions among cardiomyocyte-released molecules.

HMGB1 as well as S100A8 and S100A9 facilitates assembling of mixed TLR4 and RAGE multimers at the cell surface leading to cooperative intracellular signaling (Rouhiainen *et al*, 2013). Independence of S100A1-mediated signaling from RAGE might further contribute to differential activation of intracellular pathways, although *in vivo* involvement of RAGE through heterodimerization of S100A1 with RAGE ligands (e.g., S100B) and TLR-mediated RAGE assembly cannot be ruled out categorically. Combined use of genetically manipulated CFs and various chemical inhibitors eventually linked TLR4-induced MAPK/SAPK and p65 activation to immunomodulatory and anti-fibrotic actions of S100A1. Their activation is in line with previously reported TLR4-MyD88 endolysosomal signaling but involvement of further adaptor proteins such as interleukin adaptor-associated kinases (IRAKs) needs to be tested in further studies in order to complete the downstream signaling pathway (Mann, 2011).

Moreover, it is important to address the aforementioned time and dose dependency exerted by extracellular S100A1. The estimated half maximal effective concentration (EC50) for *in vitro* MAPK/SAPK and p65 activation of extracellular S100A1 in CFs was calculated to range between 300–600 nM. This matches interstitial S100A1 concentrations supposed to be found immediately after ischemic cardiomyocyte damage *in vivo*. S100A1 serum concentrations, even at peak levels, were below effective *in vitro* concentrations. Extracellular S100A1 actions might therefore be locally restricted to the heart and CFs, respectively, but future studies are needed to determine potential systemic effects.

Due to the complexity of the healing cascade beyond the induction of inflammatory mediators, we finally sought to assess potential *in vivo* relevance of our data by direct intramyocardial S100A1 protein injections and antibody-mediated neutralization of ischemia-released S100A1. The former yielded congruent *in vivo* expression changes with respect to selected genes. Indicating *in vivo* efficacy of extracellular S100A1, we tested its impact on MI size as clinical endpoint in terminally differentiated myocardium. Using an anti-S100A1 antibody that was shown to prevent S100A1 internalization in CFs *in vitro*, we observed pro-inflammatory and pro-fibrotic effects in mice with neutralized S100A1 in experimental myocardial infarction, mirroring our *in vitro* results. Analysis of gene expression at time points from 3 h to 7 days post-I/R further suggests that lack of immunomodulatory and anti-fibrotic TLR4-dependent signaling by S100A1 might exaggerate and prolong post-MI inflammation entailing enlarged MI size and worsened cardiac function. Albeit

$TLR4^{-/-}$ mice were protected against post-MI damage (Riad *et al*, 2008), our data indicate nonetheless that compartmentalized TLR4 signaling in CFs could convey beneficial effects.

Of note, HMBG1 inhibition in a similar setting of cardiac I/R injury in mice decreased MI size and improved post-MI functional outcome, indicating a detrimental effect on MI healing (Andrassy *et al*, 2008). Given concomitant release of numerous cardiomyocyte endogenous molecules upon ischemic damage, it is tempting to speculate that CFs and other yet to define S100A1-target cells might integrate anti- and pro-inflammatory actions as well as anti- and pro-fibrotic signaling imposed by different types of alarmins. The net effect might determine quality of the cytokine and chemokine response that directs strength and timing of subsequent cellular-mediated innate immune mechanisms (Nian *et al*, 2004).

Due to the focus of our study on early events after myocardial damage, several limitations are noteworthy. Detailed investigation of S100A1's potential impact on chronic myocardial remodeling and survival is clearly needed but was beyond the aim of the current study. Modulation of innate immune system activation by extracellular S100A1 will also be the subject of a future study. In addition, the potentially opposing effect on concurrently released pro-inflammatory and pro-fibrotic alarmins awaits clarification. It further remains to be determined whether S100A1's role as cardiac damage signaling molecule is contained to the heart or evokes an additional systemic response.

Overall, our study is the first that links S100A1 release from damaged cardiomyocytes to post-MI inflammation and healing. S100A1 targets CFs and induces an immunomodulatory and anti-fibrotic phenotype transition, providing a novel mechanism by which S100A1 could beneficially modulate myocardial wound healing. The transient release of S100A1 in patients with acute MI suggests only temporary actions of extracellular S100A1 in the heart, and continuing studies are needed to identify further targets.

# Materials and Methods

A detailed description of all methods, procedures, and information about patients and mouse models can be found in the Supplementary Methods. All animal experiments were performed according to the protocols approved by the *Institutional Animal Care and Use Committee* (Thomas Jefferson University) and complied with the *Guide for the Care and Use of Laboratory Animals*. Use of human blood samples was approved by the ethics committee of Heidelberg University Medical Faculty according to the principles of the *Declaration of Helsinki*. Written informed consent was obtained from all participating patients.

### Patients

Patients presenting with acute chest pain to the internal medicine emergency department were screened. All consecutive patients that were classified as ST-segment elevation myocardial infarction were qualified for enrollment. All patients underwent coronary angiography and received a percutaneous coronary intervention (PCI). Blood samples were taken 8–12 h after clinical onset of symptoms regardless of the time point of percutaneous coronary intervention or the

exact coronary pathology. For the control group, patients presenting with non-cardiac symptoms were enrolled. Acute myocardial infarction was excluded in this group.

### Laboratory measurements

Cardiac troponin was measured on COBAS E411 using the novel high-sensitive Troponin T assay (Roche Diagnostics Ltd., Rotkreuz, Switzerland). Creatine kinase (CK) and lactate dehydrogenase (LDH) values were determined as part of the diagnostic routines established in the central laboratory unit of Heidelberg University Hospital.

### Assessment of S100A1 serum levels

Detection of S100A1 was performed by custom-made enzyme-linked immunosorbent assay (ELISA). A microtiter plate (Maxisorb, Nunc) was coated with the capture antibody (anti-S100α rabbit polyclonal, ab11428, abcam). Serum samples were added to the corresponding wells. A standard curve was included. Following incubation with detection antibody (human S100A1 affinity-purified polyclonal sheep IgG, R&D Systems), horseradish peroxidase-conjugated revealing antibody (donkey anti-sheep IgG-HRP, sc-2473, Santa Cruz Biotechnology), and TMB (3,3′,5,5′-tetramethylbenzidine) substrate (#80091, Alpha Diagnostic International), stop solution (#80100, Alpha Diagnostic International) was added. The optical density of each well was measured with a multiplate reader (Multiskan Spectrum, Thermo Fisher Scientific) at 450 nm and corrected at 570 nm.

### Mice

Male C57B/6 mice (WT) were purchased from Jackson Laboratory (Bar Harbor, ME). Receptor for advanced glycation end products-knockout mice ($RAGE^{-/-}$) were provided by Dr Bierhaus. Toll-like receptor 4-knockout mice ($TLR4^{-/-}$) and MyD88-knockout mice ($MyD88^{-/-}$) were provided by Dr Linke and Dr Kubatzky, respectively.

### Experimental myocardial infarction

Both ligation of the left anterior descending coronary artery (LAD) and ischemia/reperfusion procedure (I/R) were performed as previously described (Wang *et al*, 2009; Gao *et al*, 2010). In brief, mice were anesthetized and the heart was manually exposed through a small chest incision and a slipknot was made around the LAD 2–3 mm from its origin with a 6.0 silk suture. Sham-operated animals were subjected to the same surgical procedures except that the suture was passed under the LAD but was not tied. In the I/R group, the slipknot was released and the myocardium was subjected to reperfusion after 30 min of ischemia. For the analysis of S100A1 serum concentration, serial blood samples were taken from the left carotid artery of anesthetized animals.

### Antibody injection

Anesthetized mice were pre-treated with either a single i.p. injection of 200 μg anti-S100A1 antibody (SA5632, custom-made, and

affinity-purified, Eurogentec, Cologne, Germany) or rabbit IgG (200 μg) 6 h prior to ischemia/reperfusion (I/R). Sham animals with rabbit IgG injection served as controls. Three days after I/R, mice were sacrificed and myocardial gene expression was assessed.

### Echocardiography

*In vivo* left ventricular function was determined by echocardiography as described previously (Wang *et al*, 2009; Gao *et al*, 2010). Mice were anesthetized with 1.5% isoflurane and two-dimensional echocardiographic views of the mid-ventricular short axis were obtained at the level of the papillary muscle tips below the mitral valve (Vevo 770, VisualSonic, Toronto, Canada). LV ejection fraction (LVEF) was calculated as previously reported (Wang *et al*, 2009; Gao *et al*, 2010).

### Determination of myocardial infarct size

Myocardial infarct size was determined by Evans blue-TTC double staining as described previously (Wang *et al*, 2009; Gao *et al*, 2010). Briefly, following 48 h of reperfusion, the ligature around the coronary artery was re-tied and 0.2 ml 2% Evans blue dye was injected into the left ventricular cavity. The heart was then quickly excised, frozen with dry ice, and sliced into five 1.2-mm-thick slices that were perpendicular to the long axis of the heart. The slides were then incubated in 1% TTC for 15 min and then digitally photographed. The Evans blue-stained area, TTC-stained area, and TTC-negative staining area (infarcted myocardium) were measured using the computer-based image analyzer SigmaScan Pro 5.0 (SPSS Science).

### Immunofluorescence of heart sections

Immunofluorescence of tissue sections was performed as previously reported (Most *et al*). In brief, mouse hearts were excised 48 h after myocardial infarction, flash-frozen, and sectioned (10 μm). Sections were fixed in MeOH/acetone (1:1), blocked, and incubated with appropriate antibodies. Following incubation with corresponding secondary antibodies, all specimens were imaged at 40× using a Sensicam high-resolution camera and Streampix image software (Norpix) with the same illumination and acquisition conditions. Conversion to binary images was done using ImageJ.

### Isolation of mouse heart cardiomyocyte and non-cardiomyocyte fraction

Cells were enzymatically isolated as described below following a previously published protocol (Volkers *et al*, 2010). Briefly, mice were anesthetized using isoflurane, euthanized by excision of the heart and subsequently the ascending aorta was fixed to a perfusion cannula. The hearts were then perfused retrogradely with tyrodes solution containing collagenase type II (Worthington Inc., Lakewood, New Jersey). When digested, hearts were cut into small pieces, transferred to a tube containing tyrodes solution supplemented with BDM, 5% fetal calf serum (FCS), and processed to a suspension using a transfer pipette. The suspension was then filtered before cardiac myocytes were allowed to completely pellet and further designated as cardiomyocyte fraction. The supernatant was saved, pelleted at 10,000 *g* for 5 min, and designated as non-cardiomyocyte fraction.

### Cell isolation and culture

Cardiomyocytes were isolated from adult rats by a standard enzymatic digestion procedure and cultivated as described (Most *et al*, 2004b). Adult rat cardiac fibroblasts (CFs) were obtained from the supernatant of cardiomyocytes and immunofluorescent staining for the fibroblast-specific discoidin domain receptor 2 (DDR2) yielded more than 99% CFs (Goldsmith *et al*, 2004). Cells were used between passages 2 and 3. For the isolation of murine ear fibroblasts (MEFs), mice were sacrificed and ears were removed. Tissue was cut into small pieces and incubated with collagenase type II. Tissue suspension was filtered and cell suspension was seeded in a cell culture dish coated with gelatin. Cells were pre-incubated with the tested inhibitors for 30 min and then further incubated with S100A1 or other reagents.

### Expression and purification of recombinant human S100A1 protein

Human recombinant S100A1 protein was produced in *Escherichia coli* as described previously (Most *et al*, 2003). The purified protein contained an endotoxic activity of about 25 EU/mg as determined by the Limulus amoebocyte lysate kit (QCL-1000, BioWhittaker, Walkersville, MD). Subsequent purification applying EndoTrap removal columns (Hyglos, Bernried, Germany) yielded a reduction to approx. 1.1 EU/mg S100A1 protein. *In vitro* S100A1 protein concentrations used in this study ranged from 0.01–10 mM, thus corresponding to 0.0011–0.11 EU/mL. Coupling of S100A1 protein to tetramethyl-rhodamine (TAMRA) was carried out by Eurogentec (Cologne, Germany).

### Western blot

Western blotting was performed as previously reported (Most *et al*, 2012) to assess cardiac protein levels. Details about antibodies and corresponding dilution can be found in the Supplementary Methods section.

### Electrophoretic mobility shift assay

Electrophoretic mobility shift assay (EMSA) was performed as described previously (Bierhaus *et al*, 1995; Rudofsky *et al*, 2007). Briefly, nuclear proteins were prepared by the method of Andrews *et al* (Andrews & Faller, 1991). 10 μg of nuclear protein was mixed with radiolabeled NF-κB oligonucleotides (5′-AGT TGA GGG GAC TTT CCC AGG C-3′). Protein–DNA complexes were separated from unbound nucleotides by electrophoresis. Gels were exposed to Amersham Hyperfilms, and densitometric quantification was carried out by using LI-COR Odyssey Software. For supershift analysis, 2.5 μg of the respective NF-κB antibody (anti-p50, -p65, -cRel, -RelB, -p52) was applied to the binding reaction (NF-κB antibodies were obtained from Santa Cruz Biotechnology).

## RNA isolation, reverse transcription, and semi-quantitative real-time polymerase chain reaction (RT-PCR)

Total RNA isolation from LV tissue samples was performed applying the TRIZOL method, according to the manufacturer's protocol (Invitrogen) as previously described (Most *et al*, 2012). First-strand cDNA synthesis from 1 μg of total RNA was carried using iScript cDNA Synthesis Kit (Bio-Rad Laboratories). Semi-quantitative PCR was carried out on a MyiQ Single-Color Real-Time PCR detection system (Bio-Rad Laboratories). For SYBR Green PCR-based expression profiling, the rat inflammatory cytokines and receptors (PARN-011) and extracellular matrix and adhesion molecules (PARN-013) RT-PCR arrays from SABiosciences were employed according to the manufacturer's protocol (for a complete gene list, see www.sabiosciences.com).

## Immunofluorescence of cells

*In vitro* immunofluorescence imaging was essentially performed as previously described (Most *et al*, 2012). Cells were seeded overnight on gelatin-coated glass coverslips, fixed with 4% paraformaldehyde, and permeabilized using Triton X-100 (Sigma-Aldrich). After incubation with appropriate primary and secondary antibodies or labeled phalloidin (Invitrogen), coverslips were mounted using Vectashield medium with DAPI (Vector Laboratories). Images were obtained with an Olympus IX81 microscope.

## Life cell imaging (lysotracker, mitotracker)

Cardiac fibroblasts were seeded on cell culture dishes with glass bottom (Fluoro Dish, World Precision Instruments) overnight. After cells were starved for 24 h with DMEM containing 0.5% FCS, media were changed to DMEM containing 50 nM lysotracker or mitotracker (Invitrogen). Immediately thereafter, rhodamine-conjugated S100A1 was added (final well concentration 1 μM). After incubation for 30 min, cells were washed 3 times with PBS and covered with 1 mL PBS. Images were obtained immediately using an Olympus IX81 microscope.

## Proximity ligation assay (Duolink®)

The Duolink assay was purchased from Olink Bioscience and performed according to the manufacturer's instructions as previously reported (Most *et al*, 2012). Samples were mounted overnight (Vectashield with DAPI, Vector Laboratories) and subsequently imaged (Olympus IX81).

## Gelatin zymography

Measurement of matrix metalloproteinase (MMP) activity was essentially performed as described previously (Siwik *et al*, 2000; Xie *et al*, 2004). In brief, unconditioned cell culture medium was mixed with sample buffer and directly loaded onto polyacrylamide gels polymerized with 0.1% gelatin as substrate. After electrophoresis under non-reducing conditions, gels were incubated with renaturating buffer for 30 min followed by overnight incubation in developing buffer. Subsequently, gels were stained in Coomassie Blue, scanned and densitometric quantification of unstained, digested

regions representing MMP activity was assessed by using LI-COR Odyssey software. All reagents required for gelatin zymography were purchased from Invitrogen.

## Interleukin-10 ELISA

Cell culture supernatant concentration of interleukin-10 was measured by enzyme-linked immunosorbent assay according to the manufacturer's protocols (R1000, R&D Systems).

## [³H]-proline incorporation assay

Measurement of total collagen synthesis was assessed by [³H]-proline incorporation assay as described in detail elsewhere (Brilla *et al*, 1994). Briefly, cardiac fibroblasts were exposed to control (vehicle) or stimulated conditions (0.1, 1, and 10 μM S100A1 or 20% FBS) for 24–72 h in the presence of 4 μCi/mL [³H]-proline and 50 μg/mL ascorbic acid. Conditioned media proteins were precipitated with an equal volume of 12% TCA. The TCA-precipitated proteins were centrifuged at 1,000 *g* for 10 min. The resulting pellets were solubilized in 0.2 N NaOH. Aliquots from each sample were counted in a Beckman scintillation counter. The remainder samples were adjusted to contain (in mM) 100 NaCl, 50 HEPES, and 3 $CaCl_2$ at pH 7.0. Collagenase type III (100 U/mL) was then added to each sample, followed by incubation for 16 h at room temperature. After collagenase digestion, the proteins were again precipitated as described above, solubilized in 0.2 N NaOH, and subjected to liquid scintillation counting. Collagenase-sensitive [³H]-proline incorporation was calculated as the difference between TCA precipitable counts before and after collagenase digestion.

## [³H]-thymidine incorporation assay

[³H]-thymidine incorporation assay was carried out as described recently (Most *et al*, 2012). Briefly, CFs were grown to 20–30% confluency in 6-well plates in DMEM containing 20% FBS and serum-starved for 24 h. CFs were then exposed to control or stimulated conditions (0.1, 1, and 10 μM S100A1 or 20% FBS) in DMEM containing 0.5% FBS for 24–48 h. [³H]-thymidine (1 μCi/mL, specific activity 20 Ci/mmol) was added in the last 4 h of incubation. For the assessment of [³H]-thymidine incorporation, media were removed at the end of incubation and cells were washed with 10% trichloroacetic acid (TCA) and digested with 0.5 N NaOH. Radioactivity in the cell digest was counted in a Beckman scintillation counter. [³H]-thymidine incorporation is expressed as total counts per minute per well.

## Proliferation assay

For the detection of proliferation rates, cardiac fibroblasts (passage 2) were trypsinized and automatically counted in suspension using a Cellometer Auto T4 (Nexcelom Bioscience, Lawrence, MA). 500,000 cells were then seeded into each well of 6-well plates and S100A1 or FCS was added to the cell culture media in the appropriate groups. Following 24, 48, or 72 h of incubation, cells were again trypsinized and automatically counted. Each experiment was carried out in quadruplicates; data from *n* = 5 individual experiments are presented as mean ± SEM.

    

## Assessment of reactive oxygen species (ROS)

Cardiac fibroblasts were loaded with the ROS-sensitive fluorogenic probe 2′,7′-dichlorodihydrofluorescein diacetate (DCFH-DA) (C6827, Molecular Probes) in DMEM containing 0.5% FBS. Cells were then washed in PBS and cultured for further 30 min to allow complete deesterification of the dye. Fluorescence measurement was carried out using an inverse Olympus microscope (IX81) equipped with U-MWU and U-MNIB filter cubes and connected to a monochromator (Polychrome II, TILL Photonics, Gräfelfing, Germany). Excitation was performed at 485 nm, and emission was detected at 535 nm. Data were analyzed with TILLVision software (TILL Photonics).

## Intracellular Ca$^{2+}$ transients

Measurement of intracellular Ca$^{2+}$ transients followed a previously published protocol (Volkers *et al*, 2010). Adult rat cardiomyocytes were loaded with Fura2-AM. Measurements were carried out using an inverse Olympus microscope (IX81) with a UV filter connected to a monochromator (Polychrome II, TILL Photonics). Baseline data from 10 consecutive steady-state transients after 15 min of electrical stimulation were averaged for the analysis of transient amplitude. Per group, cells from 5 different animals (approximately 30 cells per animal) were measured and data were pooled for analysis.

## Statistical analysis

*In vivo* results were tested for normal distribution using the Kolmogorov–Smirnov test and are presented either as mean ± SEM or as box plot (box: median with 25$^{th}$ and 75$^{th}$ percentiles, whiskers: from/to 2.5$^{th}$ and 97.5$^{th}$ percentiles). Independent samples were compared using the Mann–Whitney test. *In vitro* results are presented as mean ± SEM. Comparisons between the groups were made using unpaired Student's *t*-test. Analysis of variance was performed using the Student–Newman–Keuls method for *post hoc* analysis. MedCalc 11.1 (MedCalc, Mariakerke, Belgium) and Graph-Pad PRISM statistical software were used. All tests were two-tailed and a *P*-value < 0.05 was considered statistically significant.

**Supplementary information** for this article is available online: http://embomolmed.embopress.org

## Acknowledgements

This work was funded by a post-doctoral research grant from the Ernst und Berta Grimmke-Foundation (to DR), an Oskar-Lapp-stipend awarded by the German Cardiac Society (to DR), a Cardiology Career Development Program (CCP) stipend from the Department of Cardiology at Heidelberg University Hospital (to CS), National Institutes of Health Grants R01 HL07842 (to KP) and R01 HL092130-01 and HL092130-02S1 (to PM), a grant from the Deutsche Forschungsgemeinschaft (to PM), and grants from the German Centre for Cardiovascular Research (DZHK, to PM and HAK).

## Author contributions

DR, MB, EG, KP, HAK, and PM designed the study and experiments. DR, CS, MB, JR, ID, MV, NH, MM, and EG performed the experiments and analyzed data together with JNT, TGP, EG, KP, and PM. KFK, BL, and EG provided samples. DR and PM wrote the manuscript. JR, KFK, JNT, TGP, and BL edited the paper.

## The paper explained

### Problem

Myocardial infarction (MI)-induced cardiomyocyte death entails the coordinated activation of an acute inflammatory reaction, which is a critical prerequisite for subsequent infarct healing. However, little is known about the molecular triggers that link cardiomyocyte necrosis to initiation of the inflammatory phase. Here, we investigated the role of S100A1, the S100 isoform with highest abundance in cardiomyocytes, when released from damaged cardiomyocytes during MI.

### Results

We found that S100A1 is released from ischemic cardiomyocytes in MI patients and mice. Interstitial S100A1 is internalized by cardiac fibroblasts, leading to the activation of Toll-like receptor 4 and distinct intracellular signaling pathways, including MAP kinases and NF-κB. As a consequence, cardiac fibroblasts convert into an immuno-modulatory and anti-fibrotic phenotype. In mice, neutralization of extracellular S100A1 during MI resulted in enlarged infarct size and worsened functional cardiac performance.

### Impact

Our results hint at a novel role for extracellular S100A1 in MI patho-physiology, providing a direct link between cardiomyocyte necrosis and beneficial myocardial inflammation. Although further studies are clearly needed, this new mechanistic insight will likely impact on the development of future immunomodulatory strategies for MI patients.

## Conflict of interest

The authors declare that they have no conflict of interest.

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
