## [Review Process File · EMBO Molecular Medicine]

S100A1 is released from ischemic cardiomyocytes and signals myocardial damage via Toll-like receptor 4

David Rohde, Christoph Schön, Melanie Boerries, Ieva Didrihsone, Julia Ritterhoff, Katharina F Kubatzky, Mirko Völkers, Nicole Herzog, Mona Mähler, James N Tsoporis, Thomas G Parker, Björn Linke, Evangelos Giannitsis, Erhe Gao, Karsten Peppel, Hugo A Katus, Patrick Most

Corresponding author: Patrick Most, Section for Molecular and Translational Cardiology

Review timeline:	Submission date:	17 September 2013
	Editorial Decision:	05 November 2013
	Revision received:	17 February 2014
	Editorial Decision:	13 March 2014
	Revision received:	27 March 2014
	Accepted:	15 April 2014

Transaction Report:

Editor: Céline Carret

1st Editorial Decision

05 November 2013

Thank you for the submission of your manuscript to EMBO Molecular Medicine. We have now heard back from the three referees whom we asked to evaluate your manuscript. Although the referees find the study to be of potential interest, they also raise a number of concerns that must be addressed in a major revision of your study.

As you will see from the reports below, all three referees are concerned about the limited mechanistic insights provided by the data and all three clearly suggest experiments to improve that aspect of the work. In addition, Referees 1 and 2 are also concerned about the significance of the findings; the number of mice to be studied has to be increased and additional controls provided. In our view the suggested revisions would render the manuscript much more compelling and interesting to a broad readership. We therefore hope that you will be prepared to undertake the recommended experimental revision.

Given the potential interest of your study, we would be willing to consider a revised manuscript with the understanding that the all reviewers' concerns must be fully addressed, with additional experiments where appropriate. Please note that it is EMBO Molecular Medicine policy to allow only a single round of revision and that, as acceptance or rejection of the manuscript will depend on another round of review, your responses should be as complete as possible.

EMBO Molecular Medicine has a "scooping protection" policy, whereby similar findings that are published by others during review or revision are not a criterion for rejection. Should you decide to

submit a revised version, I do ask that you get in touch after three months if you have not completed it, to update us on the status.

I look forward to seeing a revised form of your manuscript as soon as possible.

***** Reviewer's comments *****

Referee #1 (Comments on Novelty/Model System):

There appears to be an n=1 for the mouse sample in Figure 1. I would be reluctant to draw any significant conclusions from such an assessment.

Referee #1 (Remarks):

The submitted work from Rohde describes a novel role for S100A1 as an "alarming" to signal the fibroblast cells surrounding myocytes to refrain from inducing substantial myocardial damage in the wake of ischemic injury. The concept is novel for S11A1 and brings this protein into a growing family of such paracrine factors that mediate injury responses. I enjoyed reading the manuscript and the authors have done a good job of building their case, albeit with a fairly long list of caveats, limitations, and provisos at the end of the manuscript in the Discussion section. Clearly the manuscript would be improved by redressing some of these outstanding concerns already pointed out by the authors. However, the story as it stands opens a new area of investigation with potential clinical impact and therefore should be of interest to the readers of EMBO Mol Med. To further the presentation of results without taking on significant additional studies, the following suggestions are made:

- 1) The results in Figure 1 for the mouse infarction seem to be presented on the basis of a single sample. The quality of this conclusion would certainly be enhanced by providing multiple samples and determining statistical significance as had been offered for the human samples where n=12. On a related note, if your assertion is correct that the localized production of S100A1 is much higher and diluted substantially by serum, then can you show a similar time course for infarcted hearts using a series of cardiac tissue samples to show the parallels of S100A1 production in the myocardium following infarction and demonstrate a markedly higher level of protein concentration (even better if assessed in infarct versus remote myocardium to show localization to the area of damage)?
- 2) In Figure 2, what was the time point analyzed for A and B?
- 3) In Figure 7, I would think that a potential cytoprotective effect of S100A1 would be important in myocytes as well as fibroblasts. Thus, the impact of S100A1 would not be solely limited to modulation of fibroblasts as impacting on infarct size. Even though S100A1 does not appear to be taken up by myocytes with the in vitro assay, does incubation of myocytes with S100A1 induce resistance to apoptotic cell death?
- 4) Does siRNA against TLR4 or MyD88 blunt downstream signaling in the fibroblast population as observed in Figure 6? This would help to cement the functional role of this pathway in the observed phenotype.

Referee #2 (Comments on Novelty/Model System):

This is a potentially interesting and important paper that examines the role of S100A1 as an anti-fibrotic DAMP (damage associated molecular pattern). There are several unique and potentially important aspects of this study that I think warrant publication. However many of the studies require additional control experiments and/or mechanistic studies to support the authors point of view.

Given the potential importance of this study I would favor resubmission.

Referee #2 (Remarks):

This is an interesting study that examines the role of S100A1 in modulating the phenotype of fibroblasts. There are several suggestions for the authors to consider.

1. The authors present the time course of S100A1 release from a single STEMI patient in panel 1B and then present group data in Panel 1C. Anecdotal data can be misleading. My suggestion would be either buttress Panel 1B with additional patient data or delete Panel B and simply present Panel 1C. The authors should also include a table in the Supporting Data section regarding the medications that the patients were treated with following their angioplasty, since these medications may have influenced the release of S100A1.
2. Under the section on S100A1 internalization and endocytosis by cardiac fibroblasts the authors should consider an additional control experiment, which is to stimulate the cardiac fibroblasts with LPS to see whether this increases internalization S100A1 by the cardiac fibroblasts. The reason for this is based on the author's observation that only enzymatically isolated non-cardiomyocyte fractions had uptake of S100A1 protein. Collagenase used for cell digestion contains substantial quantities of endotoxin, which may explain why the authors observe this predominately in the enzymatically isolated hearts.
3. What is the time course of S100A1 internalization and how is it affected in different genetic backgrounds including RAGE^{-/-} and TLR4^{-/-}. Some additional data to support the role of fluid endocytosis as a mechanism for internalization would strengthen the manuscript.
4. The data on S100A1 provoking an anti-fibrotic phenotype is interesting and potentially important. The authors should examine gene expression for collagen type III as well as perform a FACS analysis for alpha-smooth muscle actin staining on the cardiac fibroblasts and examine fibroblasts proliferation. If S100A1 does provoke an anti-fibrotic phenotype it may influence all types of collagen gene expression, as well as alpha-smooth muscle actin and fibroblast proliferation.
5. The S100A1 injection studies are interesting but it is unclear from the Methods what the "control" injections were. Ideally the authors should perform injections with heat denatured S100A1 as well as needle injection with saline.
6. The data with ischemia reperfusion injury and TTC staining in the mouse hearts is also potentially important finding. It would be difficult to believe that the differences in infarct area at 48 hours are from decreased fibrosis alone. The authors should present histologic data at 48 hours to determine whether the anti S100A1 antibodies also influenced myocardial inflammation and/or cell death.

Referee #3 (Remarks):

The study by David Rohde et al. first shows that S100A1 is released from damaged cardiomyocytes (CM) in vivo from patients with MI (or mice with experimental MI), and IF staining of S100A1 released from ischemic cardiomyocytes (CM) in vivo suggested to the authors that it was internalized by cardiac fibroblasts (CF). From this observation, the authors identified the molecular mechanisms involved in S100A1 internalization by CF, leading to an immunomodulatory and anti-fibrotic phenotype transition of CF. They report that extracellular S100A1 internalization by CF occurs through clathrin- and caveolin-independent fluid endocytosis not involving RAGE or TLR4. However, S100A1 endolysosomal trafficking relies on TLR4, the S100A1-TLR4-MyD88 signaling complex further activating MAPK/SAPK and NFκB pathways. Overall this study is well performed, and the data presented bring interesting information on the possible role of S100A1 on cardiac wound healing.

However, specific points need to be addressed:

1- About the origin of S100A1 in CF.

From the IF pictures shown in Fig 2B, showing that CF adjacent to murine ischemic CM *in vivo* are positive for S100A1, the authors conclude that S100A1 has been internalized by CF. To confirm this point, the authors performed *in vitro* experiments showing that if CF are subjected to supernatant from necrotic CM, they express intracellular S100A1 (Fig 2C). They conclude on a CM origin of S100A1 expressed in interstitial CF.

These *in vitro* experiments do not rule out the possibility that S100A1 expression by CF is an active process triggered by some molecules in the supernatant from hypoxic CM, such as other alarmins, HMGB1.... Did the authors characterize the molecules released from hypoxic CM? Did they test whether S100A1 neutralizing antibodies blocked S100A1 expression by CF incubated with hypoxic CM supernatant? What is the impact of endocytosis inhibitors on S100A1 expression in these experiments?

It is essential that the authors address these questions because the whole study is based on the assumption that adjacent CF passively internalize S100A1 released from ischemic CM in infarcted hearts.

2- S100A1 endolysosomal trafficking

Fig 4 shows nice IF staining of rho-S100A1 internalization in WT, RAGE^{-/-}, TLR^{-/-} and MyD88^{-/-} fibroblasts, suggesting that S100A1 endolysosomal trafficking is TLR4-dependent. Because only single cells are shown per experimental condition, one may ask whether they are representative of the whole population. For instance, is missing co-localization of S100A1 with FITC-lysotracker in TLR4^{-/-} cells occurring in all the fibroblasts? Are there alternative mechanisms of S100A1 entry and trafficking in these cells? S100A1 has been reported to bind RAGE *in vivo* in rats. It does not seem to be involved in S100A1 entry in CF. The authors should discuss this point.

3- Immunomodulatory effects of S100A1

Regarding the effects of *in vivo* injection and neutralization of S100A1 on myocardial gene expression and functional performance (Fig 7), it is shown that TGF- β 1 was highly increased following pretreatment with S100A1 antibodies of infarcted murine myocardium. What is the effect of S100A1 on the expression of this molecule? The strong induction of TGF β 1 together with the inhibition of IL-10 expression is intriguing as they are both suppressive molecules. What is the meaning in the context of MI?

1st Revision - authors' response

17 February 2014

The authors appreciate the reviewers' constructive criticism and are pleased that all three referees considered our study as innovative and potentially interesting to the readers of EMBO Molecular Medicine. The reviewers' remarks were very helpful in our attempt to improve the quality of the manuscript. In order to address the reviewers' comments, we have conducted numerous additional *in vivo* and *in vitro* experiments. Owing to the amount of new data and advanced conceptual understanding, manuscript figures and text were reorganized. Moreover, we apologize for insufficient data presentation that might have limited the comprehensibility of the manuscript in its previous version. The authors are thankful for the helpful comments and hope to have sufficiently addressed the reviewers' concerns.

Referee #1 (Comments on Novelty/Model System):

"There appears to be an n=1 for the mouse sample in Figure 1. I would be reluctant to draw any significant conclusions from such an assessment."

The authors thank Referee #1 for this valuable comment. Based on the results from controls and patients with ST-elevation myocardial infarction (STEMI), we have performed another set of experiments with sham-operated vs. LAD-ligation mice (n=10 mice in each group). Blood samples were taken 8hrs after LAD occlusion. Consistent with the human data, S100A1 serum levels were significantly elevated in the mouse LAD-ligation group (p=0.007). Overall higher S100A1 serum

levels after myocardial infarction in humans compared to mice might be explained by the greater volume of infarcted myocardium and subsequent reperfusion. In mice, a standardized permanent LAD occlusion model was used so that “wash-out” of S100A1 might have been attenuated. In comparison to the human data, basal S100A1 serum levels were relatively higher in sham-operated mice. As S100A1 is also expressed in striated muscle, the thoracic trauma caused by sham surgery might have influenced S100A1 serum levels.

These new results are now displayed in Figure 1 D. In order to clarify that the serum courses of S100A1 are based on samples from one patient and one animal, respectively, these diagrams were moved to Supplemental Figure I. They are meant to serve as representative illustrations and not as basis for experimental hypotheses. Manuscript text and Figure legends were changed accordingly.

We hope to have adequately addressed Referee #1’s important comment.

Referee #1 (Remarks):

“The submitted work from Rohde describes a novel role for S100A1 as an “alarming” to signal the fibroblast cells surrounding myocytes to refrain from inducing substantial myocardial damage in the wake of ischemic injury. The concept is novel for S100A1 and brings this protein into a growing family of such paracrine factors that mediate injury responses. I enjoyed reading the manuscript and the authors have done a good job of building their case, albeit with a fairly long list of caveats, limitations, and provisos at the end of the manuscript in the Discussion section. Clearly the manuscript would be improved by redressing some of these outstanding concerns already pointed out by the authors. However, the story as it stands opens a new area of investigation with potential clinical impact and therefore should be of interest to the readers of EMBO Mol Med. To further the presentation of results without taking on significant additional studies, the following suggestions are made:

1) The results in Figure 1 for the mouse infarction seem to be presented on the basis of a single sample. The quality of this conclusion would certainly be enhanced by providing multiple samples and determining statistical significance as had been offered for the human samples where n=12.”

The authors thank Referee #1 for this helpful remark. We hope to have sufficiently addressed this issue above (see Referee #1, Comments on Novelty/Model System).

“On a related note, if your assertion is correct that the localized production of S100A1 is much higher and diluted substantially by serum, then can you show a similar time course for infarcted hearts using a series of cardiac tissue samples to show the parallels of S100A1 production in the myocardium following infarction and demonstrate a markedly higher level of protein concentration (even better if assessed in infarct versus remote myocardium to show localization to the area of damage)?”

We are thankful for this constructive comment and agree with this reviewer that the mechanism of locally-restricted S100A1 effects due to concentration-dependent actions is of particular importance. Following the authors’ present understanding, S100A1 is passively “washed out” of damaged cardiomyocytes rather than actively produced by surrounding cells. This concept is supported by a recently published study of Bi et al. (Bi et al. Diagnostic Pathology 2013, 8: 84). Here, the authors show depletion of S100A1 in ischemic cardiomyocytes as early as 15min after LAD ligation by immunohistochemical staining.

To further confirm this notion, we have followed the reviewer’s suggestions and performed additional analyses using remote myocardium (left-ventricular posterior wall) from sham-operated and infarcted mice at different time points. As displayed in Supplemental Figure II E, the results show no difference in S100A1 content between the groups.

In order to strengthen this mechanism in the manuscript, we have changed both introduction and discussion sections and added the work of Bi et al. to the reference list (please refer to page 3, 4th paragraph, and page 8, 2nd paragraph).

“2) In Figure 2, what was the time point analysed for A and B?”

The time point of analysis for Figure 2 A and B was 48hrs after sham operation and LAD ligation, respectively. The authors regret this lack of information in the primary version of the manuscript and have adapted the legend of Figure 2 accordingly.

“3) In Figure 7, I would think that a potential cytoprotective effect of S100A1 would be important in myocytes as well as fibroblasts. Thus, the impact of S100A1 would not be solely limited to modulation of fibroblasts as impacting on infarct size. Even though S100A1 does not appear to be taken up by myocytes with the in vitro assay, does incubation of myocytes with S100A1 induce resistance to apoptotic cell death?”

The authors thank Referee #1 for this constructive proposal. We have performed additional *in vitro* experiments with isolated adult rat cardiomyocytes, testing their resistance to apoptosis induced by camptothecine (CAM) and caffeine. Pre-incubation with S100A1 had no influence on apoptotic cell death as assessed by the Cell Death Detection ELISAPlus from Roche Diagnostics, indicating that S100A1 internalization might be required for activation of a protective pathway. These new data are now displayed in Supplemental Figure VII D.

“4) Does siRNA against TLR4 or MyD88 blunt downstream signalling in the fibroblast population as observed in Figure 6? This would help to cement the functional role of this pathway in the observed phenotype.”

We are thankful for this helpful comment and agree with Referee #1 that this approach would have strengthened the role of the TLR4-MyD88-axis in S100A1-treated fibroblasts.

Indeed, we had used siRNA to knock down TLR4 and MyD88 in rat cardiac fibroblasts. Unfortunately, we have noted that fibroblasts with knock down of TLR4 or MyD88 show fundamental changes in phenotype and gene expression pattern when compared to scramble-siRNA and untreated cells, so that reactivity to extracellular stimuli (S100A1, HMGB1, LPS) was no longer reliably assessable. We have therefore switched to fibroblasts isolated from TLR4- and MyD88-knock out mice and have performed an additional set of experiments. As displayed in Figure 5 and the new version of Supplemental Figure VIII, both ERK1/2 activation and immunomodulatory and anti-fibrotic phenotype transition in response to extracellular S100A1 are blunted in both TLR4- and MyD88-knock out fibroblasts. In this way, the authors hope to have addressed the reviewer's point adequately.

Referee #2 (Comments on Novelty/Model System):

“This is a potentially interesting and important paper that examines the role of S100A1 as an anti-fibrotic DAMP (damage associated molecular pattern). There are several unique and potentially important aspects of this study that I think warrant publication. However many of the studies require additional control experiments and/or mechanistic studies to support the authors point of view. Given the potential importance of this study I would favour resubmission.”

Referee #2 (Remarks):

“This is an interesting study that examines the role of S100A1 in modulating the phenotype of fibroblasts. There are several suggestions for the authors to consider.”

1. The authors present the time course of S100A1 release from a single STEMI patient in panel 1B and then present group data in Panel 1C. Anecdotal data can be misleading. My suggestion would be either buttress Panel 1B with additional patient data or delete Panel B and simply present Panel 1C. The authors should also include a table in the Supporting Data section regarding the medications that the patients were treated with following their angioplasty, since these medications may have influenced the release of S100A1."

The authors thank Referee #2 for these constructive suggestions. We have indeed moved the serum time courses to Supplemental Figure I B-C. Instead, an additional set of mouse experiments was performed in order to illustrate the change in S100A1 serum concentration 8hrs after sham surgery vs. LAD ligation (n=10 mice in each group). Blood samples were taken 8hrs after LAD occlusion. Consistent with the human data, S100A1 serum levels were significantly elevated in the LAD-ligation group (p=0.007). Overall higher S100A1 serum levels after myocardial infarction in humans compared to mice might be explained by the greater volume of infarcted myocardium and subsequent reperfusion. In mice, a standardized permanent LAD occlusion model was used, so that "wash-out" of S100A1 might have been attenuated. In comparison to the human data, basal S100A1 serum levels were relatively higher in sham-operated mice. As S100A1 is also expressed in striated muscle, the thoracic trauma caused by sham surgery might have influenced S100A1 serum levels. These new results are now displayed in Figure 1 D.

Moreover, we have added the medications of patients in Supplemental Figure I A.

We hope to have adequately addressed Referee #2's important comment.

"2. Under the section on S100A1 internalization and endocytosis by cardiac fibroblasts the authors should consider an additional control experiment, which is to stimulate the cardiac fibroblasts with LPS to see whether this increases internalization S100A1 by the cardiac fibroblasts. The reason for this is based on the author's observation that only enzymatically isolated non-cardiomyocyte fractions had uptake of S100A1 protein. Collagenase used for cell digestion contains substantial quantities of endotoxin, which may explain why the authors observe this predominately in the enzymatically isolated hearts."

We are thankful for this excellent and important suggestion and agree with Referee #2 that extracellular co-stimulation, e.g. with endotoxin, might influence S100A1 uptake. We have therefore performed additional experiments with cardiac fibroblasts pre-stimulated with either LPS or high-mobility group box 1 protein (HMGB1) and measured S100A1 internalization. Representative Western blots are displayed in Supplemental Figure V B showing no difference between the groups. We hope to have sufficiently addressed this issue with these experiments.

"3. What is the time course of S100A1 internalization and how is it affected in different genetic backgrounds including RAGE^{-/-} and TLR4^{-/-}. Some additional data to support the role of fluid endocytosis as a mechanism for internalization would strengthen the manuscript."

S100A1 seems to be internalized in a concentration-dependent manner, which means that the amount of intracellular S100A1 not necessarily equals but correlates with the S100A1 concentration in the cell culture medium (please refer to Supplemental Figure VI A). Interestingly, the maximum amount of endocytosed S100A1 is reached after 5min (Figure 5 A). To further address this issue, we have performed additional experiments with RAGE^{-/-} and TLR4^{-/-} cells, showing no difference in the time course of S100A1 internalization when compared with WT fibroblasts (please refer to revised Supplemental Figure V C).

Moreover, we have carried out additive experiments to further elucidate the role of fluid endocytosis as a mechanism for S100A1 internalization. As displayed in the revised version of Supplemental Figure III C, internalization of S100A1 was completely blocked in the presence of macropinocytosis-inhibitor amiloride. As amiloride also prevented lysotracker-uptake, co-staining with phalloidine was performed on paraformaldehyde-fixed fibroblasts.

We hope to have addressed the issue raised by Referee #2 adequately and to have improved these points in the manuscript

“4. The data on S100A1 provoking an anti-fibrotic phenotype is interesting and potentially important. The authors should examine gene expression for collagen type III as well as perform a FACS analysis for alpha-smooth muscle actin staining on the cardiac fibroblasts and examine fibroblasts proliferation. If S100A1 does provoke an anti-fibrotic phenotype it may influence all types of collagen gene expression, as well as alpha-smooth muscle actin and fibroblast proliferation.”

The authors appreciate Referee #2's constructive remarks. We agree that expression of collagen type III and alpha-smooth muscle actin (SMA) as well as proliferation represent key variables of fibroblast phenotype conversion. We had already presented such data in the previous version of the manuscript and apologize for making this not clear enough. As displayed in Supplemental Figure VII B, collagen type III expression is significantly reduced in response to incubation with S100A1 (0.12-fold expression vs. control, 48hrs 1 μ M S100A1, assessed by RT-PCR). Accordingly, SMA expression is also diminished as illustrated in Figure 6. Furthermore, fibroblast proliferation was measured under basal conditions (0.5% FCS) and in the presence of S100A1. As shown in Supplemental Figure VIII C, incubation with S100A1 had no influence on cardiac fibroblast proliferation. In this set of experiments, 10% FCS served as positive control. Overall, the authors hope to have addressed these issues to this reviewer's satisfaction.

“5. The S100A1 injection studies are interesting but it is unclear from the Methods what the "control" injections were. Ideally the authors should perform injections with heat denatured S100A1 as well as needle injection with saline.”

The authors regret the lack of clarity in the Methods section and have added an additional paragraph for the intramyocardial injections (please refer to the Supplementary Information file, page 2, 4th paragraph). In order to expose both groups to the same experimental trauma, the control group has received saline injections. Heat denatured S100A1 was shown to have no effect on target cells *in vitro* (Supplemental Figure VI C). In accordance with the animal committee, the authors have tried to limit animal experiments to a minimum. Therefore no intramyocardial injections with inactivated S100A1 were performed. The authors still hope that this approach is acceptable for Referee #2.

“6. The data with ischemia reperfusion injury and TTC staining in the mouse hearts is also potentially important finding. It would be difficult to believe that the differences in infarct area at 48 hours are from decreased fibrosis alone. The authors should present histologic data at 48 hours to determine whether the anti S100A1 antibodies also influenced myocardial inflammation and/or cell death.”

The issue emphasized by Referee #2 here is indeed of special interest. The authors share the opinion that the differences in infarct area size cannot be explained with anti-fibrotic effects of S100A1 alone. In order to assess the time course of myocardial inflammation in those groups (sham operation with IgG control injection, I/R with IgG control injection, I/R with anti-S100A1 injection), we decided to perform additional experiments and to analyse the myocardial expression levels of inflammatory genes at different time points (3hrs to 7 days, n=4 animals in each group). Such data are now displayed in the new Supplemental Figure X, indicating prolonged and potentially unfavourable myocardial inflammation if extracellular S100A1 is neutralized. From these results the authors conclude that immediate immunomodulatory effects of S100A1 are very likely to influence the course of cardiac wound healing. S100A1 might thereby contribute to the differences in infarct area size. However, we believe that subsequent studies will have to further address this notion by measurement of inflammatory cell infiltration and spatial distribution of cardiomyocyte death in the infarct border zone. Overall, the authors hope to have sufficiently addressed this issue in the manuscript.

Referee #3 (Remarks):

“The study by David Rohde et al. first shows that S100A1 is released from damaged cardiomyocytes (CM) in vivo from patients with MI (or mice with experimental MI), and IF staining of S100A1 released from ischemic cardiomyocytes (CM) in vivo suggested to the authors that it was internalized by cardiac fibroblasts (CF). From this observation, the authors identified the molecular mechanisms involved in S100A1 internalization by CF, leading to an immunomodulatory and anti-fibrotic phenotype transition of CF. They report that extracellular S100A1 internalization by CF occurs through clathrin- and caveolin-independent fluid endocytosis not involving RAGE or TLR4. However, S100A1 endolysosomal trafficking relies on TLR4, the S100A1-TLR4-MyD88 signalling complex further activating MAPK/SAPK and NFκB pathways. Overall this study is well performed, and the data presented bring interesting information on the possible role of S100A1 on cardiac wound healing.

However, specific points need to be addressed:

1- About the origin of S100A1 in CF.

From the IF pictures shown in Fig 2B, showing that CF adjacent to murine ischemic CM in vivo are positive for S100A1, the authors conclude that S100A1 has been internalized by CF. To confirm this point, the authors performed in vitro experiments showing that if CF are subjected to supernatant from necrotic CM, they express intracellular S100A1 (Fig 2C). They conclude on a CM origin of S100A1 expressed in interstitial CF.

These in vitro experiments do not rule out the possibility that S100A1 expression by CF is an active process triggered by some molecules in the supernatant from hypoxic CM, such as other alarmins, HMGB1.... Did the authors characterize the molecules released from hypoxic CM? Did they test whether S100A1 neutralizing antibodies blocked S100A1 expression by CF incubated with hypoxic CM supernatant? What is the impact of endocytosis inhibitors on S100A1 expression in these experiments?

It is essential that the authors address these questions because the whole study is based on the assumption that adjacent CF passively internalize S100A1 released from ischemic CM in infarcted hearts.”

The authors are thankful for Referee #3’s constructive critique. We agree with Referee #3 that the issue discussed here is of special importance for the mechanistic understand of S100A1 actions in myocardial infarction. Additional experiments with cardiac fibroblasts and cardiomyocytes were performed in order to address the question if S100A1 in fibroblasts originates exclusively from damaged cardiomyocytes.

First, as proposed by this reviewer, adult cardiac fibroblasts were stimulated with LPS and HMGB1 and subsequently analysed for S100A1 expression on mRNA and protein level. Samples from adult cardiomyocytes served as positive controls. In this set of experiments, fibroblasts from all three groups did not express S100A1 (untreated, LPS-treated, HMGB1-treated). These results are now displayed in the revised version of Supplemental Figure V A.

Second, a possible influence of LPS and HMGB1 on the amount of internalized S100A1 was investigated. Here, no difference could be observed between untreated and LPS- and HMGB1-pretreated cardiac fibroblasts (please refer to revised Supplemental Figure V B).

The authors are aware of the fact that these novel *in vitro* results cannot be used to rule out other modalities of S100A1 internalization and expression in the ischemic myocardium. However, we hope to have adequately strengthened the notion of S100A1 release from damaged cardiomyocytes and subsequent passive internalization by cardiac fibroblasts in this study.

“2- S100A1 endolysosomal trafficking

Fig 4 shows nice IF staining of rho-S100A1 internalization in WT, RAGE^{-/-}, TLR^{-/-} and MyD88^{-/-} fibroblasts, suggesting that S100A1 endolysosomal trafficking is TLR4-dependent. Because only

single cells are shown per experimental condition, one may ask whether they are representative of the whole population. For instance, is missing co-localization of S100A1 with FITC-lysotracker in TLR4^{-/-} cells occurring in all the fibroblasts? Are there alternative mechanisms of S100A1 entry and trafficking in these cells? S100A1 has been reported to bind RAGE in vivo in rats. It does not seem to be involved in S100A1 entry in CF. The authors should discuss this point.”

The authors appreciate Referee #3's critical concerns about S100A1 endolysosomal trafficking. Concurrently, we apologize for the lack of convincing data due to insufficient data presentation. We have supplemented the manuscript with additional immunofluorescence images and statistical evaluations of S100A1-lysotracker co-localization in WT, RAGE^{-/-}, TLR4^{-/-} and MyD88^{-/-} cells (please refer to revised Supplemental Figure IV A and C).

In order to further investigate the role of fluid endocytosis as a mechanism for S100A1 internalization, additional experiments were performed. Co-incubation with the macropinocytosis-inhibitor amiloride resulted in absence of S100A1 uptake by cardiac fibroblasts (Supplemental Figure III C). Interestingly, application of Rac GTPase inhibitor EHT1864 lead to normal internalization but missing lysotracker co-localization in WT fibroblasts. Thus, EHT1864-treatment of WT cells imitated the situation observed in TLR4^{-/-} cells (Supplemental Figure IV B). Since activated Rac1 has been reported to implicate actin reorganization during macropinocytosis (reviewed in Lim and Gleeson, Immunology and Cell Biology 2011, 89: 836-843), this novel finding strongly suggests an involvement of Rac1-dependent cytoskeleton remodelling in S100A1 endolysosomal trafficking.

Moreover, we have tried to extensively analyse a potential involvement of RAGE in myocardial infarction-related S100A1 actions. First, both internalization and endolysosomal trafficking of S100A1 were comparable in WT and RAGE^{-/-} fibroblasts *in vitro* (please refer to Figure 4 A). Second, we could not observe any differences between WT and RAGE^{-/-} cells in terms of S100A1-mediated signal transduction and gene expression (Figure 5 E and Supplemental Figure VIII A). However, we cannot exclude an involvement of RAGE in S100A1 actions *in vivo*, as both heterodimerization of S100A1 with RAGE ligands (e.g. S100B) and TLR-mediated RAGE-assembly could occur in the milieu of necrotic cardiomyocytes. In order to emphasize a potential involvement of RAGE in S100A1-mediated gene expression, infarct size and functional performance *in vivo*, we have revised the relevant paragraph to the Discussion section (please refer to page 9, 3rd paragraph).

“3- Immunomodulatory effects of S100A1

Regarding the effects of in vivo injection and neutralization of S100A1 on myocardial gene expression and functional performance (Fig 7), it is shown that TGF- β 1 was highly increased following pretreatment with S100A1 antibodies of infarcted murine myocardium. What is the effect of S100A1 on the expression of this molecule? The strong induction of TGF β together with the inhibition of IL-10 expression is intriguing as they are both suppressive molecules. What is the meaning in the context of MI?”

The authors appreciate this relevant remark. We have investigated both TGF-b1 and IL-10 expression in cardiac fibroblasts in response to S100A1 on mRNA and protein level. Consistent with the results from anti-S100A1-injection, incubation with S100A1 resulted in markedly decreased levels of TGF-b1 together with heightened IL-10 production (please refer to revised Supplemental Figure VII B). However, the “net effect” of S100A1-mediated TGF-b1 decrease and IL-10 increase on the level of ischemia-related inflammation remains hypothetical, although the S100A1-mediated cytokine pattern seems to be beneficial with respect to infarct size and functional performance *in vivo*. Of note, expression of several pro-inflammatory cytokines like IL-1 seems to be clearly enhanced by S100A1 at the same time. That is why the authors had decided to term S100A1 actions “immunomodulatory” rather than pro- or anti-inflammatory. We have tried to improve this point in the Discussion section and have therefore rewritten the 1st paragraph on page 9. In doing so, the authors hope to have addressed the issue raised by this reviewer sufficiently.

2nd Editorial Decision

13 March 2014

Thank you for the submission of your revised manuscript to EMBO Molecular Medicine. We have now received the enclosed reports from the referees that were asked to re-assess it. As you will see the reviewers are now supportive and I am pleased to inform you that we will be able to accept your manuscript pending final editorial amendments.

Please submit your revised manuscript within two weeks.

I look forward to receiving your revised manuscript.

***** Reviewer's comments *****

Referee #2 (Remarks):

The authors have responded to the suggestions given

Referee #3 (Remarks):

The authors replied properly to the reviewers' comments and revised accordingly their manuscript. The quality of this manuscript is highly improved and it appears suitable for publication in EMM.